# E2LLM: Encoder Elongated Large Language Models for Long-Context Understanding and Reasoning

## Abstract

In the realm of Large Language Models (LLMs), the ability to process long contexts is increasingly crucial for tasks such as multi-round dialogues, code generation, and document summarization. This paper addresses the challenges of achieving high long-context performance, low computational complexity, and compatibility with pretrained models – collectively termed the "impossible triangle". We introduce E2LLM (Encoder Elongated Large Language Models), a novel approach that effectively navigates this paradox. The method involves splitting long contexts into chunks, compressing each into soft prompts via a pretrained text encoder, and utilizing an adapter to align these representations with a decoder-only LLM. To further enhance the LLM's understanding and reasoning capabilities regarding the soft prompts, we implement two training objectives: one focused on reconstructing the encoder output and the other on long-context instruction fine-tuning. Extensive experiments including Needle in a Haystack and LongBench reveal that E2LLM not only outperforms eight existing state-of-the-art (SOTA) methods across various long-context tasks, but also achieves the lowest inference time and memory usage. *Code will be available upon publication.*

## 1 Introduction

Understanding and reasoning about long context has become essential for LLMs, especially for tasks like multi-round dialogues (Bai et al., 2024a), (multi-)repository code generation (Zhang et al., 2023), and (multi-)document summarization (Giorgi et al., 2023) and question answering (Singh et al., 2021). These tasks often require processing thousands or even millions of tokens to ensure coherence and accuracy. In addition, to boost the performance of LLMs, techniques that effectively prompt LLMs to activate the domain-specific knowledge—such as chain-of-thought reasoning (Wei et al., 2022), in-context learning (Dong et al., 2022), and retrieving relevant documents or historical conversations (Ding et al., 2024b)—are also pushing the demand for longer context window.

Considerable efforts have been and are still being put into developing models that can increase the context length of LLMs, aiming at achieving strong performance for longer contexts (**T1**), while reducing the training and inference complexity (**T2**), and at the same time being compatible with pretrained models (**T3**). Achieving this compatibility is crucial for effectively leveraging the pretrained knowledge contained in these models, allowing for **parameter and sample efficiency** without necessitating extensive additional training with large datasets. However, achieving all three targets simultaneously presents a formidable challenge that often leads to some compromises, a phenomenon we refer to as the "impossible triangle", as illustrated in Figure 1.

Currently, research in this field has primarily focused on three main avenues: **modifying position embeddings, attention mechanisms, and the long input sequence itself**. The first group of methods, known as **length extension**, involves adjusting the position embeddings of LLMs to accommodate longer context extensions. This typically involves selecting a large base value for RoPE (Su et al., 2024) and then continuing pretraining or fine-tuning on the target length. While these methods effectively extend the length of LLMs with minimal model changes (**T1&T3**), they typically incur substantial computational costs during both training and inference (**T2**). For instance, even with the ability to extend context window to 2M, as seen in LongRoPE (Ding et al., 2024a), enormous resources are required to train and deploy the model, and inference times can be prohibitively long

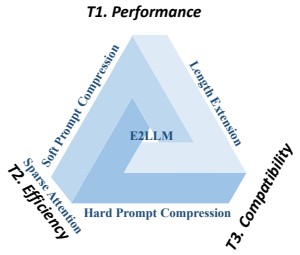

Figure 1: E2LLM solves all the challenges of the impossible triangle at the same time, namely Performance, Efficiency and Compatibility.

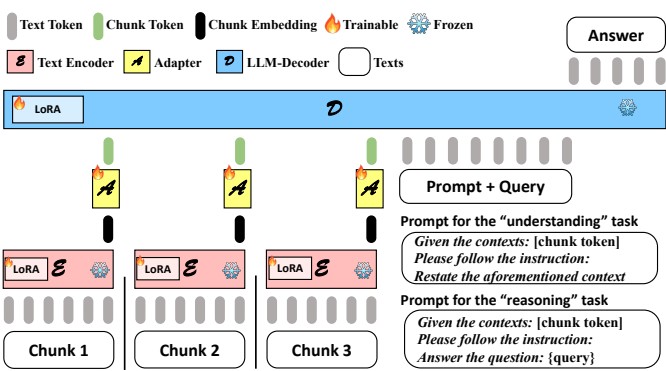

Figure 2: The E2LLM architecture.

for extended sequences. As opposed to the first group, the second one, dubbed **sparse attention**, replaces full attention in LLMs with local attention or a combination of global and local attention. This approach significantly reduces the quadratic complexity associated with full attention, even achieving linear complexity in theory (**T2**). However, a notable concern with sparse attention is its potential to neglect informative history, as certain tokens may not be attended to during the attention calculations (**T1**). Moreover, since LLMs are not originally pretrained with sparse attention, adapting them to sparse attention may require extensive training or fine-tuning (**T3**). Different from the previous two groups that change the LLMs, the third group of strategies directly compresses the input sequence to reduce its length (**T2**), which can be further divided into two subcategories. The first subgroup, known as **hard prompt compression**—exemplified by methods such as Retrieval-Augmented Generation (RAG) (Ding et al., 2024b) and LLMLingua (Jiang et al., 2023a)—tends to process compression and inference in a two-step manner. As a result, any loss of information or introduction of irrelevant content during the compression stage may adversely affect performance in the subsequent inference step (**T1**). Alternatively, the second subgroup considers **soft prompt compression**, which summarizes long contexts into embedding vectors (Chevalier et al., 2023; Tan et al., 2024). However, utilizing LLMs in these approaches to directly generate sentence-level embeddings diverges from their original pretraining objective of next token prediction. Consequently, achieving satisfactory performance in this context often demands rigorous training or fine-tuning to align the model's capabilities with the new objective (**T3**).

In this paper, we propose a novel compression based method named E2LLM (**E**ncoder **E**longated **L**arge **L**anguage **M**odels) that adeptly navigates the complexities of the "impossible triangle". Specifically, as shown in Figure 2, our method first splits a long context into chunks and compresses each chunk into an embedding vector using a pre-trained text encoder (e.g., BERT (Kenton & Toutanova, 2019)). Then, an adapter aligns the encoder's output with the input embedding space of a decoder-only LLM, such that the LLM can understand the embedding vectors resulting from the encoder. Finally, we set up two training objectives to align the encoder and decoder, including reconstructing the input text encoded by the encoder ("understanding") and long-context instruction fine-tuning ("reasoning"). We postulate that LLMs are inherently rich in knowledge; thus, properly compressed soft prompts (or the embedding vectors) can succinctly convey adequate information for LLMs to generate accurate answers. Moreover, since pre-trained encoder models are inherently crafted to produce sentence embeddings, this design allows E2LLM to capitalize on both pre-trained encoders and decoders, minimizing the requirement for extensive additional training (**T3**). Additionally, compressing each original chunk into a vector (i.e., a single chunk token) not only enhances training and inference efficiency (**T2**) but also scales up the context length significantly (**T1**). Indeed, the theoretical context window equals the product of the sequence lengths of the encoder and the decoder. The experimental results provide compelling evidence of E2LLM's superior performance in long-context scenarios, demonstrating our method's efficacy in maintaining a delicate balance between performance, efficiency, and compatibility.

To summarize, the main contributions of our work are:

- We propose E2LLM, a novel long-context modeling framework built on pretrained text encoders and decoder-only LLMs, effectively addressing the challenges posed by the "impossible triangle".

- We introduce two training objectives, including reconstructing the soft prompt given by the encoder and the long-context instruction fine-tuning, enabling the LLM to understand the soft prompt while reasoning about accurate outputs for long inputs.
- Comprehensive experiments conducted on diverse tasks and datasets demonstrate the efficiency and practicality of our proposed model and reveal its superiority over eight SOTA baselines.

## 2 RELATED WORKS

As aforementioned, prevalent methods can be categorized into three groups: modifying the position embedding (i.e., length extension), the attention mechanism (i.e., sparse attention), and the input sequence (i.e., prompt compression).

**Length Extension**: Training LLMs on sequences with limited maximum sequence lengths while ensuring generalization for longer sequences is challenging. To address this, positional extrapolation and interpolation methods have been proposed. Positional extrapolation extends positional encoding beyond the training length; for instance, ALiBi (Press et al., 2021) enhances attention with linear biases that adjust scores based on the distance between key and query positions. Instead, xPOS (Sun et al., 2023) utilizes relative position embeddings for better attention resolution and extended lengths. Another approach, CLEX (Chen et al., 2024a), replaces manual design with learned scaling factors through neural differential equations, effectively overcoming the limitations inherent in traditional positional extrapolation techniques. Positional interpolation, on the other hand, scales down input position indices and expands context windows to maintain performance across longer sequences. For example, Chen et al. (2023a) applies linear interpolation to RoPE to align maximum position indices with pre-training constraints. NTK interpolation (bloc97., 2023) modifies the base of RoPE to adjust the rotational velocity of its dimensions. To combine the strengths of these approaches, YaRN (Peng et al., 2023) merges linear and NTK interpolation with a ramp function and temperature factor, mitigating distribution shifts in the attention matrix with longer inputs. LongRoPE (Ding et al., 2024a) further enhances performance by exploiting two forms of non-uniformities in RoPE positional embedding via an efficient evolutionary search. Besides modifying position embeddings, length extension can also be achieved by employing external memory for long contexts. CEPE (Yen et al., 2024) adheres to the original Transformer architecture, using an encoder to process lengthy contexts chunk by chunk. The embeddings of tokens within each chunk given by the encoder are subsequently fed into the LLM through trainable cross-attention layers.

Despite these advancements, most approaches require continual pre-training or fine-tuning to achieve the desired length, thus entailing a considerable training burden. Additionally, inference on these extended models can be slow due to the quadratic complexity of full attention. In contrast, the proposed E2LLM does not alter the original LLM's length but compresses the input sequence into chunks of embedding vectors. This allows E2LLM to maintain the efficiency of the original LLM during both training and inference.

**Sparse Attention**: This category of methods aims to decrease the inference complexity of LLMs by manipulating attention mechanisms with novel attention masks, enabling these models to handle longer sequences. StreamingLLM (Xiao et al., 2024) demonstrates that focusing on the beginning of the sequence and the most recent tokens within a defined window (i.e., local attention) during inference maintains performance while significantly reducing computational costs to a linear scale. However, these training-free methods often fall short in various scenarios (Anagnostidis et al., 2023; Lou et al., 2024), as they may neglect informative tokens situated in the middle of the sequence. To improve performance, LM-Infinite (Han et al., 2024) reintroduces top-k tokens from the middle, but this approach necessitates the computation of all attention scores, thereby increasing computational demands. As a solution, Lou et al. (2024) propose SparseK attention, which employs an additional scoring network to assess the importance of each key-value pair and select the top-k pairs. Alternatively, LongLoRA (Chen et al., 2023a) utilizes shifted sparse attention (a variant of local attention) and fine-tunes LLMs with LoRA (Hu et al., 2021) to adapt to this mechanism. Unfortunately, as noted by (Tan et al., 2024), there remains a significant gap between sparse and full attention, which complicates the fine-tuning of pre-trained LLMs to new attention paradigms. In contrast, the E2LLM approach summarizes long-context input into soft prompt vectors, thereby reducing context length without altering the full attention mechanism in LLMs.

**Prompt Compression**: Prompt compression enhances the efficiency of LLM input processing by either condensing lengthy prompts (hard prompt compression) or learning compact prompt represen-

tations (soft prompt compression). Hard prompt compression techniques include RAG (Ding et al., 2024b), LLMlingua (Jiang et al., 2023a), Selective-Context (Li, 2023), and LongLLMLingua (Jiang et al., 2023b). RAG optimizes input by retrieving only the passages relevant to the query, while LLMlingua and Selective-Context focus on compressing extensive context without referencing the query. LongLLMLingua integrates these strategies by utilizing question-aware coarse-to-fine compression to enhance performance. However, these methods separate compression and inference into distinct steps, leading to potential error propagation that degrades performance. In contrast, E2LLM is trained end-to-end, effectively mitigating the above issue.

Soft prompt compression, proposed by Mu et al. (2023) and Ge et al. (2023), involves training LLMs to distill prompts into a more concise set of tokens that encapsulate the original prompt's knowledge for future use. Chevalier et al. (2023) extend this by developing AutoCompressor, which converts longer textual contexts into summary vectors that serve as soft prompts, which expands the LLM's context window and reduces computational costs, as examplified in LLoCO (Tan et al., 2024). However, directly using LLMs to generate sentence-level embeddings diverges from their original objective of next-token prediction. As a result, achieving satisfactory performance in this context often requires extensive training or fine-tuning to align the model with the new objective. To overcome this problem, our E2LLM leverages a pretrained sentence embedding model to represent prompts, aligning with the original training objectives of embedding models. Additionally, we note that, concurrently with our work, FocusLLM (Li et al., 2024b) has also adopted a strategy of chunking long contexts and summarizing each chunk using the hidden states of the local context from all layers of an LLM. These hidden states are concatenated to serve as the key-value cache for the same LLM, providing answers to user queries. From the perspective of E2LLM, FocusLLM essentially employs an LLM as a text encoder, which influences both training and inference efficiency.

## 3   OUR APPROACH: E2LLM

In this section, we detail the proposed E2LLM framework for understanding and reasoning over long contexts, which effectively combines the strengths of pretrained text encoders and LLM decoders.

### 3.1   MODEL ARCHITECTURE

Figure 2 illustrates the architecture of the E2LLM framework, which comprises four key components: a Chunker, a Text Encoder $\mathcal{E}_\theta$, an Adapter $\mathcal{A}_\phi$, and an LLM Decoder $\mathcal{D}_\eta$. Here, $\theta$, $\phi$, and $\eta$ denote the (learnable) parameters specific to each component. For long input contexts, E2LLM first performs chunking. Each resulting chunk is then processed by the encoder, which captures its semantics. The adapter facilitates the mapping of the encoder's outputs into the LLM decoder's embedding space, allowing the decoder to interpret these representations effectively. Ultimately, the decoder utilizes these embeddings as substitutes for the original context and executes two fine-tuning tasks—"understanding" and "reasoning"—to train the entire framework. It is essential to note that the choice of models for the encoder and decoder, the method of chunking, and the network architecture of the adapter can be customized to meet the needs of different domains. E2LLM serves as a flexible framework, seamlessly integrating these components to effectively manage long contexts while being capable of leveraging the power of more advanced components when available. We will now introduce each component in detail, following the data flow during inference in E2LLM.

**Chunker**: The Chunker is responsible for dividing long contexts into smaller, manageable chunks while ensuring that the token length of each chunk does not exceed the maximum sequence length of the text encoder. Similar to RAG, the choice of chunking strategy can impact the overall performance of E2LLM to some extent. Here, we adopt a straightforward yet effective approach: we first define a chunk size, extract the initial chunk, and then backtrack within this chunk to locate breakpoints, such as periods or line breaks. Following this, we begin a new chunk at the end of the previous one and apply the backtracking method again. We repeat this process until all text is chunked. This method helps to maintain the semantic integrity of the original texts. Note that other methods such as introducing overlap between chunks can also benefit E2LLM. Additionally, our experiments in Appendix H.2 indicate that the size of the chunks can influence this performance. Including excessive context within a single chunk can degrade performance. This occurs primarily because a high compression ratio may render the embedding vector too generic, compromising specificity. Conversely, an excessively small chunk size can disrupt the semantic integrity of sen-

tences, which can also negatively affect performance. Furthermore, we highlight that the impact of the chunker in E2LLM is less pronounced when aligning the encoder and decoder, as introduced in the sequel. In contrast to RAG, where the retriever (encoder) and the generator (decoder) are two distinct models without alignment, E2LLM benefits from this cohesion. This alignment minimizes the risk of inconsistency in text interpretation, which can arise when models are pretrained on different corpora and objectives. More discussions on chunk size is provided in Appendix I.2.

**Text Encoder** $\mathcal{E}$: After chunking, we input each chunk into the text encoder to generate the corresponding embedding vector. Notably, most pretrained encoders, such as GTE (Li et al., 2023) and BGE (Xiao et al., 2023), are trained via contrastive learning. This means the [CLS] token (the dark gray token in Figure 2), which serves as the embedding vector, typically captures only the discriminative information necessary for differentiating between chunks, while information essential for the LLM decoder to answer the query may be discarded. To mitigate this limitation, we adopt low-rank adaptation (LoRA) (Hu et al., 2021) to make text encoder trainable during the alignment process. This allows the encoder to extract information from the original text within the chunks that is beneficial for the LLM's performance.

**Adapter** $\mathcal{A}$: To facilitate the LLM's understanding of the chunk-wise semantics derived from the encoder's output, we employ an Adapter to map the encoder's output into the input embedding of the LLM. Since the hidden dimensions of the text encoder and the LLM decoder may differ, the Adapter is a vital component. Specifically, we utilize a two-layer Multi-Layer Perceptron (MLP) with the GELU activation function (Hendrycks & Gimpel, 2016) as the adapter network. This Adapter is applied to each chunk embedding individually, and we refer to its output as the *chunk token*, *soft prompt*, or *summary vector*, which are then processed by the subsequent LLM. The Adapter is initialized randomly and trained from scratch during the alignment phase.

**LLM Decoder** $\mathcal{D}$: Finally, we concatenate the chunk tokens (the green tokens in Figure 2) and the text tokens corresponding to the prompt and query, and ask the LLM to generate the answer for the query. In our experiments, we select Llama2 (Touvron et al., 2023) as the LLM Decoder due to its popularity in both academic research and industry applications. Additionally, we employ LoRA to fine-tune the Decoder as part of the alignment process between the encoder and decoder.

## 3.2 TRAINING TASKS

Now we focus on training the the adapter as well as the LoRA branch of the encoder and the decoder to enhance the E2LLM's ability to comprehend lengthy input contexts and effectively reason about the corresponding answers. To accomplish this, we introduce two distinct training tasks.

The first task is designed to improve the LLM's understanding of the input. As depicted in Figure 2, once the LLM receives chunk tokens from the adapter, we prompt it to restate or reconstruct the input. We refer to this as the "understanding" task. The specific prompt used is "Given the contexts: [chunk token]\n Please follow the instruction:\nRestate the aforementioned context". Notably, this task is self-supervised, allowing us to curate a significant amount of training data to ensure that the LLM comprehensively grasps the embeddings provided by the adapter. However, in our experiments, we utilize only the input from long-context instruction fine-tuning data for this task. Given that these inputs are often too lengthy to be fully reconstructed at once, we employ a sliding window approach, reconstructing the original context in segments based on a few consecutive chunks until the entire input has been restated.

On the other hand, the second training task enables the LLM to generate answers based on the chunk tokens (i.e., the long context) and the user's query. We refer to this as the "reasoning" task, and the prompt crafted for this purpose is "Given the contexts: [chunk token]\n Please follow the instruction: \n Answer the question: {query}".

It is important to note that the "understanding" task serves as an auxiliary task, while our primary focus remains on the "reasoning" task. We determine the final checkpoints exclusively based on the validation loss associated with the "reasoning" task. In this context, we do not anticipate that E2LLM can achieve lossless compression of the context. However, we believe that the LLM decoder is capable of retaining or comprehending essential information from the context. The LLM operates as a "suggestion feature" for input methods, leveraging hints to generate meaningful responses. In this case, the chunk tokens provided by the text encoder serve as these essential hints.

**Maximum Context Window**: Theoretically, the maximum sequence length of E2LLM equals the product of the encoder and decoder's sequence lengths. However, as previously mentioned, setting the chunk size to match the encoder's sequence length presents challenges, as it may hinder the encoder's ability to retain all pertinent information within a single chunk. Thus, we need to choose a proper chunk size. As a result, the practical length of E2LLM is determined to be the chunk size multiplied by the sequence length of the LLM's decoder. In actuality, we set the maximum chunk size of 512 characters, which is approximately equivalent to 100 tokens. Hence, the context length has been expanded by nearly 100 times. When using Llama2-7B as the decoder with a sequence length of 4,000 tokens, the final context window of E2LLM reaches approximately 400,000.

**Time and Space Complexity during Inference**: Let us denote the original context length (excluding the prompt or instruction) as $L$ and the chunk size in E2LLM as $C$. Therefore, the total number of chunks becomes $L/C$. For each chunk, the resulting time and space complexity from the text encoder is $\mathcal{O}(C^2)$. Given that there are $L/C$ chunks, the overall complexity for the encoding step is $\mathcal{O}(CL)$. In practice, since all chunks can be processed in parallel, the time complexity can be further reduced by a constant factor. Subsequently, we pass the $L/C$ chunk tokens to the LLM decoder, which yields a complexity of $\mathcal{O}(L^2/C^2)$. In summary, the total time and space complexity is $\mathcal{O}(LC + L^2/C^2)$. To substantiate the efficiency of E2LLM during inference, we conduct empirical experiments that assess both inference time and memory usage (cf. Section 4.4). Moreover, we provide a discussion on the complexity of existing SOTA methods in Appendix A.

### 3.3 RELATION TO VISION-LANGUAGE MODELS (VLMS)

E2LLM draws inspiration from recent advancements in VLMs, including mini-GPT4 (Zhu et al., 2024), LLaVA (Liu et al., 2024), Qwen-VL (Bai et al., 2023), and InternVL (Chen et al., 2024b). These VLMs utilize adapters to align pretrained vision encoders with LLM decoders, enabling the LLMs to process image tokens outputted by the vision encoders. In this framework, both the vision encoder and LLM decoder are pretrained independently, offering a flexible approach that allows for the alignment of high-performing vision and language models, thereby maximizing their capabilities. Notably, VLMs excel at performing OCR (Optical Character Recognition) (Islam et al., 2017) tasks, effectively recognizing and outputting text present within images. Motivated by the success of VLMs, we propose that by aligning text encoders (i.e., embedding models) with LLM decoders using an adapter, LLMs can similarly interpret sentences encoded by the text encoders and draw inferences based on this comprehension. Furthermore, as both the encoder and decoder in our approach operate within the same modality, we anticipate that the alignment process will be more straightforward than that required for models functioning across different modalities, potentially reducing the amount of data needed for alignment. Conversely, the reconstruction task employed in training E2LLM is self-supervised, enabling us to amass a vast dataset of text to enhance the LLM's contextual understanding. In contrast, the alignment task in VLMs relies on supervised image-text pairs, which are notably more challenging to collect.

## 4 EXPERIMENTS

In this section, we first evaluate the performance of E2LLM across three key tasks: document QA, document summarization, and Needle-in-a-Haystack retrieval. We then broaden our assessment to include its performance on LongBench. Additionally, we examine the training and inference efficiency of E2LLM and conduct a series of ablation studies to gain further insights.

For comparison, we benchmark E2LLM against eight baselines. These include length extension techniques (YaRN (Peng et al., 2023) and CEPE (Yen et al., 2024)), sparse attention strategies (StreamingLLM (Xiao et al., 2024) and LongLoRA (Chen et al., 2023b)), as well as hard and soft prompt compression methods (RAG (Gao et al., 2024), LongLLMLingua (Jiang et al., 2023b) and LLoCO (Tan et al., 2024)). All baseline methods are built upon the same foundational model, Llama2-7B, with the original Llama2-7B included as an additional baseline. We refer readers to Appendix B for a brief overview of all baselines before delving into the experimental results.

### 4.1 DOCUMENT SUMMARIZATION AND QUESTION ANSWERING (QA)

We utilize two datasets for summarization—QMSum and GovReport—and three datasets for QA—Quality, NarrativeQA, and TriviaQA. Detailed information about these datasets can be found in Appendix C and Table 4. It is noted that Quality and TriviaQA feature shorter lengths compared to

Table 1: Performance on Long-Context datasets. The best results are in **bold**, the second are underlined, and the third are wavy underlined.

| Methods | Trainable Parameters | Context Window | Extension Method | QMSum | | GovReport | | Quality | | NarrativeQA | | TriviaQA | |
|---|---|---|---|---|---|---|---|---|---|---|---|---|---|
| | | | | G-mean↑ | PPL↓ | G-mean↑ | PPL↓ | F1↑ | PPL↓ | F1↑ | PPL↓ | F1↑ | PPL↓ |
| Llama2-7B | 0M | 4K | - | 11.51 | 84.92 | 5.50 | 9.04 | 9.38 | 1,688.10 | 4.65 | 2,111.23 | 12.06 | 1,956.51 |
| StreamingLLM | 0M | 4M | Sparse Attn. | 3.62 | 220.12 | 4.51 | 330.54 | 2.00 | 230.72 | OOM | OOM | 14.53 | 596.87 |
| LongLoRA | 140M | 100K | Sparse Attn. | 8.98 | 14.48 | 16.35 | 2.88 | 7.65 | 381.32 | OOM | OOM | 19.69 | 438.25 |
| CEPE | 1.31B | 128k | Len. Exten. | 10.77 | 154.16 | 4.82 | 52.32 | 2.33 | 1,192.35 | OOM | OOM | - | - |
| YaRN | 17M | 64K | Len. Exten. | 12.31 | 16.22 | 6.72 | 2.94 | 13.80 | 31.32 | OOM | OOM | 20.22 | 106.43 |
| RAG | 0M | +∞ | Hard Comp. | 7.24 | 19.11 | 3.89 | 4.97 | 10.36 | 131.50 | 2.77 | 59.43 | 16.40 | 111.26 |
| LongLLMLingua | 0M | 40K | Hard Comp. | 8.93 | 17.55 | 4.56 | 23.53 | 10.89 | 51.91 | 4.53 | 31.36 | 14.01 | 76.06 |
| LLoCO | 17M | 128K | Soft Comp. | 12.99 | 46.32 | 5.73 | 6.42 | **14.37** | 9.44 | 10.87 | 16.88 | **63.21** | **10.80** |
| E2LLM | 16M | 400K | Soft Comp. | **14.61** | **13.68** | 18.78 | **2.75** | 12.94 | 7.94 | 12.35 | 13.31 | 33.37 | 12.69 |

*  For complete experimental results with more metrics, please refer to the experimental results details section in the Appendix C.

the summarization datasets, while NarrativeQA is notably longer. For our experiments, we employ the validation sets of each dataset for testing and split the training sets into training and validation subsets using a 95:5 ratio. For summarization tasks, we evaluate all methods using the Rouge metric, which compares n-grams of the generated text to those of the reference text, focusing on Rouge-1, Rouge-2, and Rouge-L for various levels of token overlap. The overall performance is represented through the geometric mean (G-mean) of these values, with higher scores signifying better quality in generated summaries. In contrast, for Document QA, we adopt the method from (Shaham et al., 2023) to measure unigram overlap between generated and reference answers while normalizing whitespace, lowercasing, and removing stopwords and punctuation. Precision and recall are then calculated based on the number of unigram tokens, and the overall performance is assessed using the F1 score as a token-level metric. Additionally, we compute the perplexity (PPL) of the correct answer across all datasets as a semantic-level metric, gauging how well a model predicts the given answer. The results for all baseline methods are presented in Table 5. Following this, we now discuss the outcomes for each category of methods in detail.

**Soft prompt compression**: It is apparent that the proposed **E2LLM consistently achieves either the best performance or ranks within the top three across all nine evaluated methods**. We further show in Appendix C that the performance of E2LLM is insensitive to the context length. The other soft prompt compression technique, LLoCO, also demonstrates commendable performance, especially in QA tasks, highlighting the effectiveness of soft prompt compression techniques. However, LLoCO's performance declines slightly in summarization tasks, which aligns with observations in its original publication (see Table 1 in (Tan et al., 2024)). LLoCO leverages AutoCompressor (Chevalier et al., 2023) as its text encoder, operating without additional training. AutoCompressor utilizes Llama2 to generate summary vectors for each chunk, designed to retain only the information necessary for subsequent chunks while discarding other potentially valuable content, as highlighted by (Rau et al., 2024). In QA tasks, only the relevant portions of the long context are required to prompt the LLM for accurate answers, aligning well with AutoCompressor's training objectives. In contrast, summarization tasks necessitate an overall understanding of the entire context. Consequently, since the summary vectors produced by AutoCompressor do not encapsulate all information within each chunk, LLoCO's performance in summarization is adversely affected. Unlike LLoCO, E2LLM can train its encoder to be readily adapted for diverse purposes.

**Hard prompt compression**: Similar to LLoCO, the hard prompt compression method LongLLM-Lingua also excels in Document QA compared to summarization. The challenge of compressing long context into 3,000 non-consecutive tokens manifests in two significant ways: (i) the chosen token count is insufficient for summarizing the full long context; and (ii) the non-consecutiveness can hinder LLM comprehension, potentially leading to inaccurate answers. Additionally, the performance of this method is sensitive to hyperparameters, such as the chunk or passage size, which is crucial when selecting relevant passages for the query prior to token selection. These issues are also prevalent in RAG. Further complicating matters, the bi-encoder utilized in RAG may not retrieve relevant passages as effectively as the cross-encoder employed in LongLLMLingua. Inconsistencies can also arise when the retriever (encoder) and the generator (decoder) interpret the same text, as they are pretrained on different corpora (Li et al., 2024a; Ding et al., 2024b). E2LLM addresses these

issues by aligning the encoder and decoder through the adapter, which provides a global semantic embedding for each chunk and allows the decoder to utilize all chunks as inputs. This approach differs from selectively choosing some tokens from each chunk, enabling E2LLM to effectively retain relevant information and consistently surpass both hard prompt compression methods.

**Sparse attention**: On the flip side, the sparse attention method LongLoRA shows superior performance on summarization tasks but struggles with QA tasks. This disparity can be attributed to the shift shot attention mechanism utilized in LongLoRA, which allows for overlapping attention blocks and enhances global information flow—an essential aspect of summarization requiring a holistic view of all tokens. Nevertheless, the sparse attention mask limits information flow between two arbitrary tokens. Consequently, when relevant parts of the long context are inaccessible during Document QA, LongLoRA may fail to deliver accurate answers due to the loss of vital contextual information. StreamingLLM is training-free and implements a $\Lambda$-shaped attention mask that further limits overall information flow. Without training, models initially designed with full attention struggle to adapt to this mask, diminishing their performance across all datasets. E2LLM addresses these challenges by employing the original full attention mask rather than resorting to sparse attention while effectively compressing passages into soft prompts (i.e., semantic summaries). This strategy enables E2LLM to consistently achieve superior performance compared to sparse attention methods.

**Length extension**: Lastly, we observe that the length extension method, YaRN, strikes a balance between QA and summarization, generally finishing third best across all tasks and metrics. Like E2LLM, it encompasses all relevant information; however, as noted in previous research (Chen et al., 2023a), attention mechanisms can become dispersed in exceedingly long contexts, diffusing focus across numerous token positions and achieving performance inferior to E2LLM. CEPE faces a similar challenge. Moreover, training the cross-attention layers in CEPE usually requires a vast amount of data (around 20 billion tokens, as suggested in (Yen et al., 2024)). This need arises because these layers are absent from the original language model (LLM). In our experiments, the number of tokens for each task is less than 0.1 billion, raising concerns that the cross-attention layers may not be sufficiently trained. Thus, integrating cross-attention layers into existing LLMs may pose compatibility issues without access to a substantial dataset for re-training. Additionally, CEPE operates within a pretraining framework in which the encoder processes a fixed-length segment of the sequence initially. This segment is then used to predict the remainder of the sequence for the decoder, effectively functioning as a text completion task. Notably, for TriviaQA, the context length is often shorter than the encoder's predefined length, leaving the decoder without any input. This results in the decoder producing irrelevant answers after training on the TriviaQA data. In contrast, E2LLM addresses the issue of attention dispersion encountered by length extension methods by not extending the decoder's length. Instead, it trains the decoder to interpret the soft prompts generated by the encoder, thereby enhancing performance.

### 4.2 NEEDLE IN A HAYSTACK

The Needle-in-a-Haystack benchmark is a framework designed to assess models' abilities to pinpoint specific information embedded within extensive text. In this context, the term "needle" refers to a precise fact or statement concealed within a lengthy "haystack" of text, while "depth" denotes the needle's position within that context, measured from the beginning. A depth of 100% indicates that the needle is situated very close to the answer. For comparison, we have selected five representative methods from various categories: the original Llama2-7B, YaRN, LongLoRA, LongLLMlingua, and LLoCO. The methods that require training utilize data collected from all five tasks discussed in the previous subsection. For detailed experimental settings, please refer to Appendix D. Below, we present the results of all evaluated methods.

As shown in Figure 3, **the proposed E2LLM outperforms all other methods, achieving an overall score that is 17% higher than the second-best method, YaRN**. Importantly, E2LLM is insensitive to both the length of the context and the depth of the needle, as it treats all context chunks equally. In contrast, while YaRN ranks second, its recall accuracy declines significantly when the context length exceeds 4,000 tokens, indicating that its recall capability is limited to the original context. Compared to the original Llama2-7B, YaRN consistently enhances recall across various context lengths, but Llama2-7B's training context length limitation hinders its ability to retrieve needles over 4,000 tokens from the retrieval question. On the other hand, sparse attention methods, like LongLoRA, struggle in recall-intensive tasks, as informative tokens may be masked by sparse attention mech-

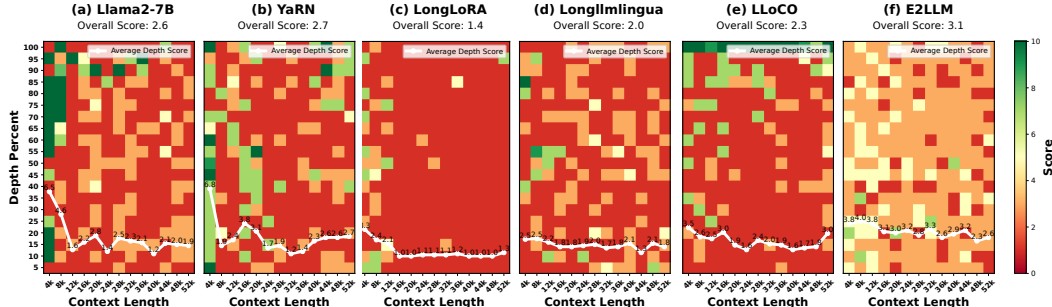

Figure 3: Pressure test on Needle in a Haystack conducted at 13 lengths (4k to 52k) across 20 depth percentage ranges (5% to 100%). The average depth score represents the mean score across the depth axis for each length.

Table 2: Performance on LongBench Benchmark. The best results are in **bold**, the second are underlined, and the third are ~~wavy underlined~~.

| Methods | Single-Document | | | Multi-Document | | | Summarization | | | Few-shot | | | Synthetic | | Code | |
|---|---|---|---|---|---|---|---|---|---|---|---|---|---|---|---|---|
| | NQA | QAS | MFQA | HQA | WQA | MSQ | GOVR | QM | MN | TREC | TQA | SAM | PC | PR | LCC | RBP |
| LLama2-7B | 8.36 | 11.96 | 25.82 | 16.67 | 13.83 | 8.36 | 10.51 | 1.85 | 14.87 | 25.11 | 51.97 | 17.24 | 0.11 | 0.03 | **48.59** | 10.70 |
| StreamingLLM | 0.25 | 7.21 | 8.05 | 6.46 | 5.79 | 4.23 | 3.03 | 2.11 | 6.09 | 1.02 | 20.82 | 2.56 | 0.12 | 0.24 | 6.97 | 4.23 |
| LongLoRA | 10.59 | 16.27 | 26.17 | 26.33 | 21.49 | 15.37 | 10.71 | 9.30 | 9.74 | 37.00 | 33.97 | 9.68 | 4.01 | 4.52 | 30.36 | 28.34 |
| CEPE | 2.69 | 4.74 | 10.96 | 7.12 | 6.10 | 5.11 | 11.23 | 7.86 | 8.67 | 0.24 | 21.02 | 9.53 | 1.55 | 0.07 | 18.84 | 19.58 |
| YaRN | **16.45** | 18.03 | 27.90 | 27.86 | 24.32 | 17.17 | 7.18 | 10.02 | 6.84 | 39.00 | 55.67 | 17.61 | 3.42 | 2.92 | 39.29 | **45.12** |
| RAG | 5.37 | 9.12 | 20.73 | 20.70 | 15.23 | 8.53 | 3.83 | 12.83 | 3.97 | **62.25** | 62.28 | 21.46 | 4.25 | 43.08 | 12.75 | 21.31 |
| LongLLMLingua | 6.09 | 11.65 | 25.73 | 14.38 | 8.03 | 4.81 | 14.33 | 9.06 | 13.44 | 11.0 | 8.75 | 7.24 | 3.67 | 2.00 | 14.26 | 18.04 |
| LLoCO | 13.37 | 20.60 | 18.99 | 37.73 | 24.68 | 15.94 | 2.21 | 11.46 | 9.75 | 19.00 | 86.38 | 16.75 | 7.37 | 0.55 | 37.73 | 18.06 |
| E2LLM | 12.78 | **21.94** | 16.77 | 26.45 | 25.51 | 12.43 | 14.55 | **19.06** | **15.85** | 1.52 | 83.96 | 25.86 | 4.50 | 4.96 | 27.43 | 24.47 |

anisms, resulting in poorer outcomes. Akin to E2LLM, LongLLMLingua also exhibits robustness against variations in context length and needle depth due to their flexible information retrieval before generating an answer. However, its effectiveness is sensitive to hyperparameter settings, such as chunk size; if chunk sizes greatly exceed the needle length, irrelevant information may obscure the needle, leading to retrieval failures. E2LLM mitigates this issue by processing all chunks in the decoder. Finally, LLoCO is limited in retrieving needles only when they are proximity to the answer. This limitation stems from the nature of LLoCO's encoder, AutoCompressor, which generates summary vectors primarily aimed at predicting the next chunk. As the inserted needle often bears little relation to the adjacent chunk, it may be filtered out, impairing overall performance.

## 4.3 LONGBENCH

We further conduct a comprehensive evaluation of various methods using LongBench (Bai et al., 2024b), which encompasses all major long-text application areas, including single-document QA, multi-document QA, summarization, few-shot learning, synthetic tasks, and code completion. In particular, Within the few-shot learning category, we assess TREC for question classification, TQA (i.e., TriviaQA) for reading comprehension, and SAM for conversation summarization. The synthetic task category includes Passage Retrieval (PR) constructed from English Wikipedia and Passage Count (PC), which aims to determine the number of unique passages within a given set. For code completion category, we evaluate LCC for long code completion and RepoBench-P (RBP) for repository-level code completion. More details are provided in Appendix E. Importantly, we do not train the models with new data; instead, we utilize the checkpoints obtained from the previous subsection. The results for LongBench are summarized in Table 2.

Again, **E2LLM achieves the best results across all baselines**, securing the top rank in six tasks, the second rank in four tasks, and the third rank in two tasks. In comparison, Yarn trails behind E2LLM, ranking first in five tasks, second in three tasks, and third in two tasks. It is noteworthy that while we train all models solely with natural language data, Yarn generalizes well to code data, yielding strong results for LCC and RBP. However, Yarn's high training and inference complexity may limit its practical applicability (see Figure 4). LLoCO ranks third overall, achieving first place in three tasks, second in four tasks, and third in one task. This reinforces the notion that soft prompt

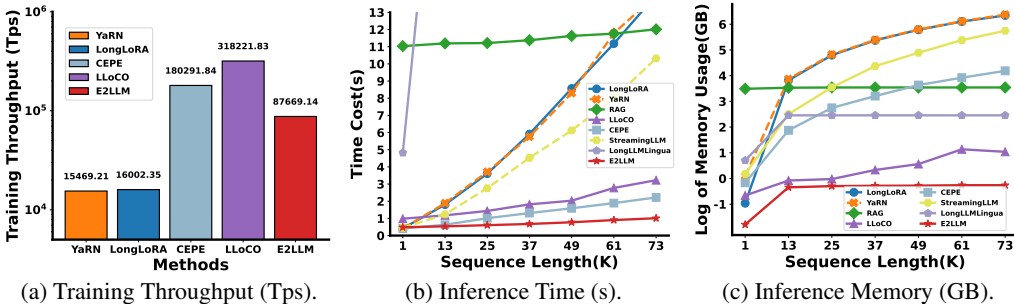

(a) Training Throughput (Tps).  (b) Inference Time (s).  (c) Inference Memory (GB).

Figure 4: Comparison of all methods on training and inference efficiency.

compression approaches are promising for various long-context understanding and reasoning tasks. E2LLM's superiority over LLoCO can be attributed to its flexible and lightweight design.

### 4.4 TRAINING AND INFERENCE EFFICIENCY

We only present the conclusions here due to the page limit; further discussions are in Appendix G.

**Training Throughput**: We evaluate the training throughput of several methods requiring training, including YaRN, LongLoRa, CEPE, LLoCO, and E2LLM. As shown in Figure 4a, CEPE, LLoCO, and E2LLM exhibit significantly higher training throughput compared to YaRN and LongLoRA. Notably, LLoCO prepares the summary vectors or soft prompts offline, allowing it to fine-tune only the LLM decoder. In contrast, CEPE's training of cross-attention layers scales linearly with context length, which contributes to its superior performance.

**Inference Time and Memory**: As displayed in Figures 4b and 4c, **E2LLM stands out with impressive results, demonstrating the lowest runtime and memory usage, especially for lengthy sequences at 73K**. This efficiency is primarily due to its relatively high compression ratio of approximately 100 times, which dramatically reduces the number of chunk tokens processed by the LLM decoder to a size much smaller than the original number of text tokens. In comparison, LLoCO achieves a compression ratio of 32 times. Additionally, we observe that the runtime behavior of all methods aligns with the theoretical time complexity outlined in Table 3.

### 4.5 ABLATION STUDY

Due to page limitations, we present only the conclusions here, with detailed results and discussions available in Appendix H. (i) It is essential to employ the "understanding" loss and to train both the encoder and decoder using LoRA. (ii) Incorporating overlap between chunks also proves beneficial for E2LLM. (iii) **E2LLM can benefit from more powerful encoders and decoders**, indicating that newly developed open-source models could further enhance its performance. (iv) We check E2LLM's sensitivity to hyperparameters, including the weight of the "understanding" loss, the rank of LoRA for both the encoder and decoder, and the number of MLP layers in the adapter. Each of these factors has an optimal value in practice. (v) Finally, we examine the impact of chunk size on performance. Results presented in Table 11 indicate that performance metrics exhibit relatively small differences across the various chunk sizes tested in this study. This suggests that the alignment process in E2LLM effectively mitigates the influence of chunk size on performance. However, selecting an optimal chunk size can still lead to a slight performance improvement.

### 5 CONCLUSION

In this paper, we present E2LLM, a novel approach to address the challenges of enhancing long-context performance in LLMs. It effectively navigates the "impossible triangle" by strategically splitting long contexts into chunks, compressing them into embedding vectors, and utilizing an adapter to align these representations with a decoder-only LLM. Two training objectives are employed to facilitate the understanding of soft prompts by the LLMs, resulting in superior performance in long-context scenarios. Experimental findings reveal that E2LLM effectively outperforms existing approaches in balancing the long-context performance, computational efficiency, and model compatibility. We believe that E2LLM offers a flexible framework for aligning text encoders and LLM decoders, with considerable potential for enhancement as more powerful encoders and decoders become available.

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

Table 3: Time and space complexity of various methods.

| Methods | Time Complexity | Space Complexity |
|---|---|---|
| Llama2-7B | $\mathcal{O}(L^2)$ | $\mathcal{O}(L^2)$ |
| StreamingLLM | $\mathcal{O}(L(M+N))$ | $\mathcal{O}(L(M+N))$ |
| LongLoRA | $\mathcal{O}(L^2)$ | $\mathcal{O}(L^2)$ |
| CEPE | $\mathcal{O}(L(C/\tau + 1/2) + L^2/4)$ | $\mathcal{O}(L(C+1/2) + L^2/4)$ |
| YaRN | $\mathcal{O}(L^2)$ | $\mathcal{O}(L^2)$ |
| RAG | $\mathcal{O}(LC/\tau + C^2 K^2)$ | $\mathcal{O}(LC + C^2 K^2)$ |
| LongLLMLingua | $\mathcal{O}(L^2)$ | $\mathcal{O}(L^2)$ |
| LLoCO | $\mathcal{O}(LC + L^2/C^2)$ | $\mathcal{O}(LC + L^2/C^2)$ |
| E2LLM | $\mathcal{O}(LC/\tau + L^2/C^2)$ | $\mathcal{O}(LC + L^2/C^2)$ |

## A  COMPLEXITY OF EXISTING METHODS

The original Llama2-7B and YaRN rely on the quadratic time and space complexity inherent to the self-attention mechanism. In contrast, StreamingLLM modifies the attention strategy to focus solely on the initial $M$ starting tokens and $N$ recent tokens, resulting in a linear relationship between time and space complexity and the context length. Regarding LongLoRA, its inference process employs a global attention mechanism, leading to time and space requirements equivalent to those of YaRN and the original Llama2-7B. CEPE divides the context into two segments, with the initial portion processed through parallelized embedding, represented in the table by the constant $\tau$ denoting concurrency, the subsequent self-attention and cross-attention mechanisms exhibit quadratic and linear complexities, respectively. RAG involves both the embedding and retrieval processes, establishing a direct correlation with the chunk size $C$ and the number of retrieved chunks $K$. An increase in $K$ results in slower speeds and greater space consumption, albeit with improved performance. For LongLLMLingua, it incorporates question-aware coarse-grained and fine-grained compression processes, which significantly consume time and space resources during the multiple computations of perplexity. LLoCO exhibit nearly identical time complexity to E2LLM, as both involve encoding and decoding processes. However, it is important to note that while E2LLM's encoding process shares similarities with the embedding process of RAG and can be executed concurrently, LLoCO is constrained by the AutoCompressor, which operates serially and thus cannot be parallelized. Moreover, the efficiency of both methods is directly tied to $C$, E2LLM benefits from high compatibility and can utilize long-context sentence embedding models such as BGE, GTE, and Jina-embedding as encoders, while LLoCO is limited by the AutoCompressor, restricting the chunk size range to 0-1536.

## B  OVERIEW OF BASELINE METHODS

The following provides a brief overview of all baselines:

- **Llama2-7B** (Touvron et al., 2023): This refers to Llama2-7b-Chat[1] without additional training or fine-tuning, serving as the backbone for the other methods.

- **YaRN** (Peng et al., 2023): YaRN is a position interpolation method designed to effectively extend the context window of models trained with Rotary Position Embeddings (RoPE) (Su et al., 2024). This method leverages the advantages of both linear and NTK interpolation. Note that the computational complexity of YaRN is quadratic in the context length during both training and inferece. We implement a scale factor of 16 and integrate LoRA (Hu et al., 2021) into the self-attention module, utilizing a rank of 16. This results in a total of 17 million trainable parameters.

- **CEPE** (Yen et al., 2024): CEPE employs an encoder-decoder framework designed to efficiently manage long contexts by breaking them into manageable chunks. The encoder generates embeddings for each token within these chunks, which are then fed into the LLM decoder via cross-attention, in line with the original Transformer architecture. We use LLaMA-MLM-Large[2] as

---

[1] https://huggingface.co/meta-llama/Llama-2-7b-chat-hf
[2] https://huggingface.co/hyen/LLaMA-MLM-Large

the encoder, with a total of 1.31B trainable parameters. During the warm-up stage, we train the cross-attention mechanism from scratch, followed by simultaneous training of both the encoder and cross-attention in the standard training phase. It is important to note that CEPE only presents a pretraining approach where the encoder initially processes a fixed-length segment of a sequence. This processed portion is then used to predict the remainder of the sequence for the decoder, functioning as a text completion task. In instances where a sequence is shorter than the predefined length of the encoder, the decoder is not provided with any input, which limits training flexibility. Unlike traditional Transformer fine-tuning, where the prompt and response are respectively inserted into the encoder and decoder, CEPE operates differently and does not support this method.

- **StreamingLLM** (Xiao et al., 2024): These approaches are training-free and utilize a $\Lambda$-shaped sparse attention mask, allowing tokens to only attend to the beginning of the sequence and recent tokens within a defined window. In our implementation of StreamingLLM, we set the start size at 4, while the recent size was set to 2000.

- **LongLoRA** (Chen et al., 2023b): This method utilizes shifted short attention instead of full attention during training and incorporates Position Interpolation (Chen et al., 2023a) and LoRA for fine-tuning an LLM to extend its context window. During inference, it reverts to full attention rather than sparse attention. We set the LoRA rank to 16 and fine-tune the self-attention, embeddings, and normalization modules, resulting in 140M trainable parameters.

- **RAG** (Gao et al., 2024): RAG operates with two core processes: retrieval and generation. During the retrieval phase, we adopt GTE-Large-en (Li et al., 2023) as the retriever to recall the top-20 relevant context chunks, each with a maximum length of 512 characters, based on cosine similarity. These context chunks then serve as prompts for the large language model (LLM) during the generation phase. Notably, RAG is training-free, offering flexibility in its application. However, it is essential to acknowledge that the retriever and the generator are distinct models trained on different corpora and with different objectives, which may lead to inconsistent interpretations of the same text (Li et al., 2024a; Ding et al., 2024b).

- **LongLLMLingua** (Jiang et al., 2023b): This method builds upon the framework established by LLMLingua (Jiang et al., 2023a) with the goal of identifying and removing non-essential tokens from prompts. This method begins by selecting passages, denoted as $x^{\text{passage}}$, that are relevant to the user query $x^{\text{query}}$ and that maximize the conditional probability $p(x^{\text{passage}}|x^{\text{query}})$. To achieve this, it utilizes a large language model (LLM), specifically the quantized Llama-7B-GPTQ, as a cross-encoder to rank the pairwise relevance of passages. It is important to note that cross-encoders tend to be significantly more computationally demanding than the bi-encoder retriever typically employed in RAG, although they offer higher accuracy. Once the relevant passages are identified, the method proceeds to select the most pertinent tokens $x_i$ from each passage, aiming to maximize the difference in perplexity: $\text{PPL}(x_i|x_{<i}) - \text{PPL}(x_i|x^{\text{query}}, x_{<i})$. This process is also facilitated by the LLM. Ultimately, the selected tokens, limited to a total of 3000, are provided to the LLM to formulate an answer to the query. Note that the selected tokens may be non-consecutive, which can complicate the LLM's understanding of their semantic meaning.

- **LLoCO** (Tan et al., 2024): LLoCO utilizes Autocompressors (Chevalier et al., 2023) to encode long context offline into summary vectors or soft prompts. LLoCO omits the adapter used in E2LLM since its decoder is the same LLM (i.e., LLama2-7B) as in the encoder AutoCompressor. As a result, the decoder can effectively understand the summary vectors generated by AutoCompressor after being fine-tuned with LoRA. One advantage of LLoCO is that its text encoder, AutoCompressor, considers the interdependencies of long-context chunks autoregressively. However, this also presents a limitation: the long context can only be processed sequentially, one chunk after another. By contrast, E2LLM can process all chunks in parallel and is more suitable for long context. Consistent with other methods, we employ LoRA on self-attention module with a rank of 16, resulting in the number of trainable parameters to be 17M.

- **E2LLM**: For our E2LLM, we utilize the GTE-Large-en (Li et al., 2023) as the encoder, which is fine-tuned using LoRA with a rank parameter set to 8. Additionally, we utilize a two-layer MLP network with a GeLU (Hendrycks & Gimpel, 2016) activation function as the adapter. As for the decoder component, we leverage Llama2-7B-Chat, also fine-tuning it through LoRA with a rank of 8, and the final number of trainable parameters is 16M. Regarding the Adapter, its structure is designed as a two-layer MLP. The first layer's input and output neuron numbers correspond to the embedding dimensions of the encoder and decoder, respectively, with GELU used as the

Table 4: Dataset Statistics.

| Dataset | Task Type | #Train. Samp. | #Eval. Samp. | Samp. Len. |
|---------|-----------|---------------|--------------|------------|
| QMSum | Summarization | 1,257 | 272 | 14,428.78 |
| GovReport | Summarization | 10,000 | 500 | 11,204.00 |
| Quality | DocumentQA | 5,046 | 2,086 | 6,797.66 |
| NarrativeQA | DocumentQA | 3,000 | 200 | 52,158.88 |
| TriviaQA | DocumentQA | 10,000 | 500 | 1,075.90 |

activation function. The second layer maintains equal input and output dimensions, aligning with the decoder's embedding size.

## C  MORE DETAILS OF DOCUMENT SUMMARIZATION AND QA DATASETS

In order to evaluate the effectiveness of E2LLM, we leverage five publicly available datasets that encompass both Summarization and Document Question-Answering (DocumentQA) tasks. The data statistics are shown in Table 4.

- **QMSum**[3] (Zhong et al., 2021) is a newly devised, human-annotated benchmark designed for the query-based multidomain meeting summarization task. It comprises an extensive range of query-summary pairs across 232 meetings in diverse fields. Specifically, we included 1,257 training samples and used 272 samples for inference. The average length of the samples in this dataset is 14,428.78 tokens.

- **GovReport**[4] (Huang et al., 2021) contains elongated reports by the U.S. Government Accountability Offices and the Congressional Research Service, complemented by summaries and abstracts hand-written by experts, which is of the summarization task genre. For training purposes, 10,000 random samples were utilized, and for inference, 500 samples were arbitrarily selected from the validation sets. The average length of the sampled data is 11,204.00 tokens.

- **Quality**[5] (Bowman et al., 2022) is a DocumentQA dataset comprising 5,046 training samples and 2,086 inference samples with contexts that have an average length of 6,797.66 tokens. Further, we convert the original single-choice data format of the dataset into the QA format.

- **NarrativeQA**[6] (Kovcisky et al., 2018) is another DocumentQA dataset, primarily extracted from comprehensive book texts and film scripts from varied sources. The challenge here lies in generating concise answers from potentially disordered and lengthier texts. We randomly sample 3,000 pieces of data for training, while randomly choosing 200 samples for inference. The average sample length is 52,158.88 tokens.

- **TriviaQA**[7] (Joshi et al., 2017)is also a high-quality DocumentQA dataset that houses over 650K question-answer-evidence triples. It includes 95K question-answer pairings authored by trivia enthusiasts and independently sourced evidence documents. We selected 10,000 and 500 samples for training and inference respectively, with the average sample length amounting to 1,075.90 tokens.

For the task of Summarization, the performance of all methods is measured using the Rouge (Lin, 2004) metric, which operates by comparing the n-gram of the generated text with that of the reference text. Specifically, we leverage Rouge-1, Rouge-2, and Rouge-L to assess the overlap between the single-token, consecutive dual-tokens, and the longest common subsequence (LCS) in the generated text by LLM and the reference text. We also compute their geometric mean, denoted as G-mean, and higher values reflect higher quality of the generated summaries.

---

[3]https://github.com/Yale-LILY/QMSum
[4]https://huggingface.co/datasets/ccdv/govreport-summarization
[5]https://huggingface.co/datasets/emozilla/quality
[6]https://github.com/google-deepmind/narrativeqa
[7]https://huggingface.co/datasets/mandarjoshi/trivia_qa

Table 5: Performance on Long-Context datasets. The best results are in **bold**, the second are underlined, and the third are wavy underlined.

| Methods | Trainable Parameters | Context Window | QMSum | | | | GovReport | | | | Quality | | | NarrativeQA | | | TriviaQA | | |
|---|---|---|---|---|---|---|---|---|---|---|---|---|---|---|---|---|---|---|---|
| | | | R1 | R2 | RL | G-mean | R1 | R2 | RL | G-mean | Prec. | Recall | F1 | Prec. | Recall | F1 | Prec. | Recall | F1 |
| Llama2-7B | 0M | 4K | 21.90 | 4.91 | 14.21 | 11.51 | 10.68 | 2.86 | 5.46 | 5.50 | 6.16 | 25.46 | 9.38 | 3.04 | 13.52 | 4.65 | 6.72 | 76.66 | 12.06 |
| StreamingLLM | 0M | 4M | 7.59 | 1.15 | 5.43 | 3.62 | 7.46 | 3.39 | 4.76 | 4.51 | 1.50 | 5.50 | 2.00 | OOM | OOM | OOM | 8.43 | 76.99 | 14.53 |
| LongLoRA | 140M | 100K | 13.92 | 4.82 | 10.79 | 8.98 | 27.04 | 9.92 | 16.29 | 16.35 | 7.41 | 9.99 | 7.65 | OOM | OOM | OOM | 13.03 | 49.28 | 19.69 |
| CEPE | 1.31B | 128k | 19.22 | 3.66 | 17.74 | 10.77 | 10.53 | 1.08 | 9.89 | 4.82 | 1.35 | 29.89 | 2.33 | 1.41 | 21.31 | 2.19 | - | - | - |
| YaRN | 17M | 64K | 21.54 | 5.34 | 16.24 | 12.31 | 12.93 | 4.13 | 5.69 | 6.72 | 13.20 | 19.42 | 13.80 | OOM | OOM | OOM | 13.53 | 49.45 | 20.22 |
| RAG | 0M | +∞ | 11.45 | 3.32 | 10.05 | 7.24 | 8.15 | 1.75 | 4.14 | 3.89 | 5.71 | 40.17 | 10.36 | 0.83 | 3.41 | 2.77 | 8.25 | 54.35 | 16.40 |
| LongLLMLingua | 0M | 40K | 16.42 | 3.56 | 12.18 | 8.93 | 8.63 | 2.19 | 5.20 | 4.56 | 9.13 | 26.34 | 10.89 | 5.26 | 30.78 | 4.53 | 5.20 | 77.33 | 14.01 |
| LLoCO | 17M | 128K | 23.71 | 5.51 | 16.79 | 12.99 | 11.69 | 3.11 | 5.18 | 5.73 | 16.81 | 15.03 | 14.37 | 11.85 | 11.34 | 10.87 | 64.04 | 64.03 | 63.21 |
| E2LLM | 16M | 400K | 25.37 | 6.55 | 18.75 | 14.61 | 33.14 | 10.75 | 18.59 | 18.78 | 13.44 | 14.95 | 12.94 | 13.53 | 13.79 | 12.35 | 33.22 | 34.51 | 33.37 |

Table 6: Performance as a function of context length. The best results are in **bold**, the second are underlined, and the third are wavy underlined.

| Method | QMSum | | | | | | | | | | NarrativeQA | | | | | | | | | |
|---|---|---|---|---|---|---|---|---|---|---|---|---|---|---|---|---|---|---|---|---|
| Context Length | 0K-6K | | 6K-12K | | 12K-18K | | 18K-24K | | 24K+ | | 0-24K | | 24K-48K | | 48K-72K | | 72K-96K | | 96K+ | |
| Metric | G-mean | PPL | G-mean | PPL | G-mean | PPL | G-mean | PPL | G-mean | PPL | F1 | PPL | F1 | PPL | F1 | PPL | F1 | PPL | F1 | PPL |
| Llama2-7B | 13.05 | 28.57 | 11.99 | 85.35 | 11.54 | 84.31 | 12.56 | 81.74 | 10.32 | 85.60 | 3.10 | 75.81 | 10.71 | 178.28 | 7.51 | 250.81 | 0.61 | 2303.08 | 2.48 | 2215.08 |
| StreamingLLM | 3.27 | 36.35 | 4.21 | 168.63 | 3.32 | 224.24 | 3.26 | 356.17 | 2.45 | 362.41 | 4.36 | 79.34 | 2.53 | 135.71 | OOM | OOM | OOM | OOM | OOM | OOM |
| LongLoRA | 5.91 | 12.92 | 8.13 | 13.17 | 8.30 | 14.65 | 9.66 | 15.97 | 7.44 | 17.31 | 3.23 | 11.93 | 9.47 | 12.17 | OOM | OOM | OOM | OOM | OOM | OOM |
| CEPE | 11.66 | 128.01 | 10.42 | 144.34 | 9.29 | 161.28 | 8.21 | 145.54 | 6.56 | 234.24 | 3.37 | 3568.12 | 2.65 | 2272.04 | OOM | OOM | OOM | OOM | OOM | OOM |
| YaRN | 13.57 | 14.52 | 12.10 | 14.02 | 12.88 | 17.06 | 11.49 | 17.75 | 6.33 | 18.90 | 7.19 | 13.94 | 6.59 | 17.16 | OOM | OOM | OOM | OOM | OOM | OOM |
| RAG | 6.12 | 17.94 | 8.72 | 17.58 | 9.65 | 20.95 | 9.03 | 19.59 | 6.24 | 19.39 | 2.40 | 12.98 | 2.14 | 41.35 | 2.55 | 60.28 | 2.14 | 58.32 | 1.43 | 57.20 |
| LongLLMLingua | 7.73 | 11.25 | 9.83 | 15.12 | 8.72 | 16.25 | 9.08 | 19.66 | 8.87 | 21.55 | 7.84 | 26.52 | 6.23 | 29.45 | 3.16 | 29.96 | 1.72 | 38.53 | 1.03 | 48.53 |
| LLoCO | 13.63 | 34.56 | 12.78 | 41.27 | 13.15 | 47.45 | 12.13 | 47.87 | 10.03 | 56.30 | 10.89 | 13.32 | 10.67 | 15.67 | 10.88 | 17.31 | 11.42 | 16.19 | 9.43 | 18.54 |
| E2LLM | 15.04 | 12.69 | 15.27 | 13.47 | 14.14 | 13.95 | 14.26 | 13.33 | 15.31 | 13.92 | 12.12 | 13.45 | 12.41 | 12.87 | 12.76 | 12.96 | 12.23 | 13.65 | 11.97 | 13.71 |

Concerning the task of DocumentQA, we adopt the method demonstrated by (Shaham et al., 2023), which computes the unigram overlap between the generated and reference answers. This is accomplished by normalizing white-spaces, lower-casing, excluding stopwords and punctuation. Based on the number of unigram tokens, in conjunction with the token quantity of the generated and reference answers, we calculate precision, recall, and F1. Again, a higher value indicates a more precise answer by the model.

In Table 5 we present the above metrics of all methods for document summarization and QA.

Next, we investigate the sensitivity of the models' performance to variations in context length. To do this, we categorize samples from the QMSum and NarrativeQA datasets into five groups based on their context lengths and then evaluate the perplexity (PPL) of the answers within each group. Our findings are summarized in Table 6.

The results presented in the table indicate that E2LLM demonstrates a strong resilience to variations in context length for both summarization (QMSum) and question-answering (NarrativeQA) tasks, consistently achieving the best results among all models. This robustness can be attributed to the "understanding" task incorporated during the training of E2LLM (see Section 3.2). By reconstructing different parts of the context, E2LLM effectively comprehends the information, regardless of its length.

Notably, the performances of YaRN, LongLoRA, CEPE, RAG, and LongLLMLingua also exhibit insensitivity to context length. On the other hand, LLoCO's performance declines slowly with increasing context length. Finally, streamingLLM and the original Llama2-7B demonstrate sensitivity to context length; streamingLLM loses more information in the middle of the context as length increases due to its specific Λ-shaped attention mask, while Llama2-7B struggles to handle long contexts altogether, as its maximum length has not been extended.

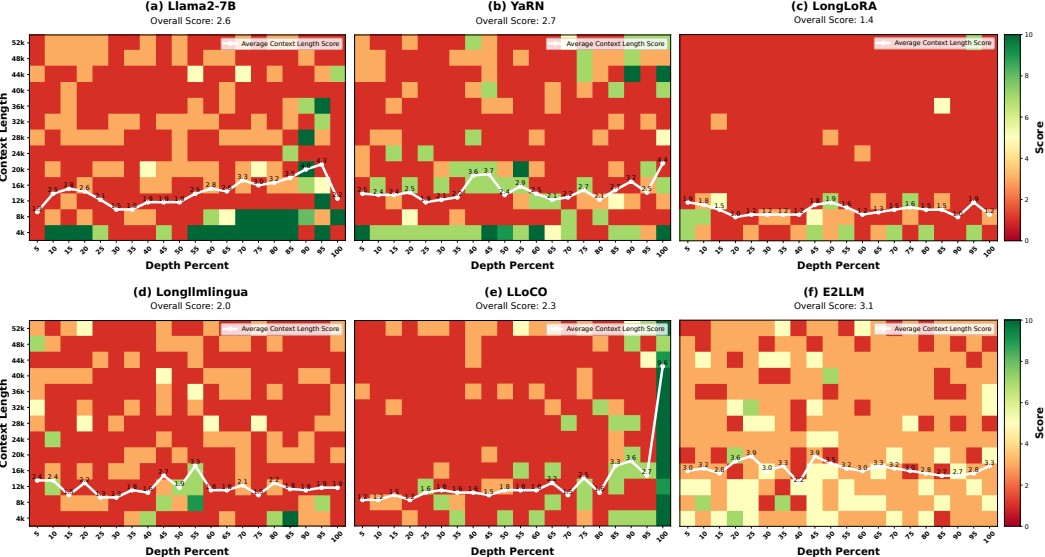

Figure 5: Score avaraged over context length as a function of depth percentage in Needle in a Haystack.

## D  MORE DETAILS OF NEEDLE IN A HAYSTACK

To assess the models' ability to retrieve information from various positions within a lengthy context, we utilize the well-established Needle in a Haystack benchmark. In this framework, a random fact or statement (referred to as the "needle") is embedded within a lengthy context (the "haystack"), and its position from the beginning of the context is termed the "depth." For our experiment, we selected 49 essays from Paul Graham's website as the haystack. The specific needle we inserted is: "The best thing to do in San Francisco is eat a sandwich and sit in Dolores Park on a sunny day," accompanied by the retrieval question: "What is the best thing to do in San Francisco?" We then task the model with retrieving this precise statement, using GPT-4o mini to evaluate performance based on predefined criteria and scoring templates. To ensure a thorough evaluation, we prepare contexts of varying lengths, ranging from 4,000 to 52,000 tokens, and examine 20 different ranges of depth percentages, from 5% to 100%. Note that a depth of 100% signifies a position that is quite close to the answer.

For comparison, we have selected five representative methods from various categories: the original Llama2-7B, YaRN, LongLoRA, LongLLMlingua, and LLoCO. The methods that require training are trained on the data collected from the five tasks outlined in Appendix C. Note that we use all training samples from QMSum, GovReport, Quality, and NarrativeQA, but randomly select 3,000 samples from TriviaQA, as the sample size of this dataset is much larger than that of others, but the average context length is the shortest. The total number of training samples is around 13,000. For the original Llama2-7B, whose sequence length is only 4,000, we truncate the long context from the left such that the truncated context length is 4,000.

In addition to the results presented in Section 4.2, we further illustrated the average score over the context length as a function of depth percentage in Figure 5. It is evident that the performances of Llama2-7B, YaRN, LongLoRA, LongLLMLingua, and E2LLM are largely insensitive to the depth at which the needle is inserted. In contrast, LLoCO achieved the best results when the needle was positioned close to the answer, as discussed at the end of Section 4.2. Furthermore, E2LLM typically delivers the best performance across all depths.

## E  DESCRIPTION OF LONGBENCH

We employ LongBench(Bai et al., 2024b) as the benchmark to evaluate the effectiveness of E2LLM and baseline models. LongBench offers a comprehensive bilingual and multi-task dataset characterized by diverse sequence lengths, distributions, patterns, languages, and domains, designed to rigorously evaluate long-context understanding capabilities. Given that our base model is Llama2-

Table 7: Data Statics for LongBench. Details of the datasets are collated by Li et al. (2024b).

| Task | Task Type | Metric | Avg. Length | Language | Sample |
|---|---|---|---|---|---|
| NQA | Single-doc QA | F1 | 18,409 | EN | 200 |
| QAS | Single-doc QA | F1 | 3,619 | EN | 200 |
| MFQA | Single-doc QA | F1 | 4,559 | EN | 150 |
| HQA | Multi-doc QA | F1 | 9,151 | EN | 200 |
| WQA | Multi-doc QA | F1 | 4,887 | EN | 200 |
| MSQ | Multi-doc QA | F1 | 11,214 | EN | 200 |
| GOVR | Summarization | Rouge-L | 8,734 | EN | 200 |
| QM | Summarization | Rouge-L | 10,614 | EN | 200 |
| MN | Summarization | Rouge-L | 2,113 | EN | 200 |
| TREC | Few shot | Accuracy | 5,177 | EN | 200 |
| TQA | Few shot | F1 | 8,209 | EN | 200 |
| SAM | Few shot | Rouge-L | 6,259 | EN | 200 |
| PC | Synthetic | Accuracy | 17,210 | EN | 200 |
| PR | Synthetic | Accuracy | 9,289 | EN | 200 |
| LCC | Code | Edit Sim | 1,235 | Python/C#/Java | 500 |
| RBP | Code | Edit Sim | 4,206 | Python/Java | 500 |

Table 8: Performance on RULER Benchmark. The best results are in **bold**, the second are underlined, and the third are wavy underlined.

| Contex Length | 4K | | | | 8K | | | | 16K | | | |
|---|---|---|---|---|---|---|---|---|---|---|---|---|
| Task | VT | CWE | FWE | QA | VT | CWE | FWE | QA | VT | CWE | FWE | QA |
| LLama2-7B | **27.00** | **85.60** | **74.33** | **63.00** | - | - | - | - | - | - | - | - |
| LongLoRA | 1.60 | 16.60 | 9.33 | 55.50 | 2.20 | 13.40 | 10.33 | **44.00** | 2.00 | 5.80 | 4.00 | **52.00** |
| YaRN | 19.80 | 15.20 | 20.33 | 57.00 | 1.80 | 10.30 | 11.67 | 34.50 | 1.40 | 3.90 | 5.33 | 29.00 |
| LongLLMLingua | 5.20 | 7.60 | 44.67 | 14.50 | **4.20** | 5.70 | **24.33** | 16.0 | **7.00** | 2.00 | **27.33** | 15.50 |
| LLoCO | 0.00 | 27.70 | 24.67 | 32.50 | 0.00 | **24.10** | 17.00 | 28.50 | 0.00 | **20.90** | 22.67 | 20.00 |
| E2LLM | 0.00 | 15.60 | 21.33 | 40.50 | 0.00 | 14.30 | 18.67 | 37.00 | 0.00 | 16.30 | 19.33 | 37.50 |

| Contex Length | 32K | | | | 64K | | | | 128K | | | |
|---|---|---|---|---|---|---|---|---|---|---|---|---|
| Task | VT | CWE | FWE | QA | VT | CWE | FWE | QA | VT | CWE | FWE | QA |
| LLama2-7B | - | - | - | - | - | - | - | - | - | - | - | - |
| LongLoRA | 0.40 | 1.80 | 1.670 | **33.50** | OOM | OOM | OOM | OOM | OOM | OOM | OOM | OOM |
| YaRN | 1.20 | 2.80 | 2.00 | 28.50 | OOM | OOM | OOM | OOM | OOM | OOM | OOM | OOM |
| LongLLMLingua | **6.20** | 0.30 | 11.33 | 18.50 | **5.20** | 0.30 | 13.33 | 15.0 | **5.20** | 0.40 | **21.67** | 4.50 |
| LLoCO | 0.00 | 0.10 | **24.00** | 4.50 | 0.00 | 2.40 | **15.67** | 9.00 | 0.00 | 3.30 | 4.33 | 2.00 |
| E2LLM | 0.00 | 3.50 | 16.67 | 28.00 | 0.00 | 4.90 | 13.33 | **16.50** | 0.00 | 2.50 | 8.67 | **7.50** |

7B, we have conducted an extensive evaluation across all 14 English tasks and 2 code tasks. More details regarding the benchmark are listed in Table 7. For methods that require training, the training data utilized are identical to those employed during the "Needle in a Haystack" experiment.

## F   RULER

In this section, we present the results of Llama2-7B, LongLoRA, YaRN, LongLLMLingua, LLoCO, and E2LLM on the RULER benchmark. RULER primarily consists of four types of tasks:

- **Retrieval**: This task involves the Needle-in-a-Haystack test, which evaluates retrieval capability using diverse types and quantities of "needles".

- **Muti-hop Tracing**: The variable tracking task (VT) serves as a minimal proxy for coreference chain resolution, examining the ability to trace entities across multi-hop connections.
- **Aggregation**: This task entails the extraction of common or frequent words (CWE and FWE), functioning as a proxy for summarization to test the ability to aggregate relevant information across long-range contexts.
- **Question Answering**: For this task, distracting information is added to the input of existing short-context QA datasets in order to assess question-answering capabilities at various context sizes.

We do not consider the retrieval tasks here, as they can be considered variants of the Needle-in-a-Haystack test. For the VT task, we set the number of variable name-binding chains and the number of times binding variable names in each chain to be 1 and 4, respectively. For the CWE and FWE tasks, we set the frequency of ten common words to be 30, uncommon words to be 3, and alpha as 2.0. Finally, for the QA task, we use two single-hop short-context QA datasets SQuAD and HotPotQA. For models that requires training, we reuse the checkpoints trained in Section 4.2.

The results are listed in Table **??**. Given the diversity of tasks presented in RULER, we can clearly identify the strengths and weaknesses of each baseline method. Although Yarn and LongLoRA perform relatively well in the QA task, they struggle significantly with the CWE and FWE tasks. This is likely due to an attention distraction problem, which hampers their ability to focus on specific common or frequent words. Additionally, both methods encounter out-of-memory issues when the context length exceeds or equals 64K; for reference, we utilized an A100 GPU with 80GB of memory for inference. This suggests that the space complexity of YaRN and LongLoRA is too high for scenarios with limited resources. On the other hand, LongLLMLingua excels in the FWE task but underperforms in the others. The soft compression methods, E2LLM and LLoCO, manage to strike a balance between performance on the aggregation (CWE and FWE) and QA tasks, yielding comparable results. E2LLM tends to favor QA tasks, while LLoCO is better suited for aggregation tasks. It is worth noting that E2LLM can take advantage of increasingly sophisticated text encoders that are continuously being open-sourced, as demonstrated in our ablation studies; meanwhile, the encoder used by LLoCO is fixed to AutoCompressor. Lastly, we observe that all methods perform poorly on the VT task, which demands a nuanced comprehension of the long context, presenting a challenge that may be too great for the current models.

## G  MORE DISCUSSIONS ON TRAINING AND INFERENCE EFFICIENCY

**Training Efficiency**: We assess the training throughput of all methods requiring training, including YaRN, LongLoRa, CEPE, LLoCO, and E2LLM. The experiments conducted on a single eight A100 GPU-equipped machine focus on measuring the number of processed tokens per second (tps), which serve as our evaluation metric. The configuration for all baselines adheres to the respective parameters specified in each of their original papers, and for our E2LLM, a chunk size of 512 characters is set.

As demonstrated in Figure 4a, YaRN is clearly the least training-efficient method due to its necessary handling of the quadratic time complexity associated with the context length, stemming from its lack of original long context compression. LongLoRA, utilizing a sparse attention mechanism, offers slightly improved efficiency compared to YaRN by eliminating the need to compute the attention between some query-key pairs. Conversely, both CEPE and LLoCO demonstrate high throughput. CEPE initially processes all chunks of the long context in a parallel way, akin to E2LLM, but retains token-level embedding opposed to chunk-level embedding. This method then only trains the cross-attention linking the encoder and decoder, introducing linear time complexity relative to the long context length. In contrast, E2LLM trains the decoder relative to the compressed context length, thus explaining CEPE's higher throughput. Surpassing these, LLoCO performs remarkably well in training efficiency given that the summary vectors or soft prompt are prepared offline ahead of time, necessitating only the fine-tuning of the LLM decoder. E2LLM finally, processes context chunks in parallel during the encoding phase and fine-tunes the decoder module efficiently with LoRA, thus also demonstrating commendable training efficiency.

**Inference Efficiency**: We now proceed to examine the inference efficiency of various methods. We begin by selecting seven differing context lengths that range from 1K to 73K; both YaRN and LongLoRA encounter out-of-memory issues at a context length of 74K. For each selected context

length, we randomly select ten samples and truncate them to their predefined lengths. Upon averaging the runtime and GPU memory costs (i.e., peak allocated memory) over these samples, we reveal the results as a function of context length in Figure 4b and 4c.

Our model, E2LLM, exhibits the most impressive performance metrics, particularly in terms of runtime and memory usage, even for lengthy sequences of up to 73K tokens. In contrast, both YaRN and LongLoRA display significantly higher resource consumption, primarily due to the quadratic complexity inherent in full attention mechanisms during inference (notably, LongLoRA employs a full attention mask at this stage). Unlike LongLoRA, StreamingLLM utilizes a $\Lambda$-shaped sparse attention mask during inference, resulting in reduced time and memory costs. However, as indicated in the official implementation, for any given context, StreamingLLM must initially load the entire KV cache associated with that context. During the subsequent generation process, it utilizes Sink Attention to preserve the KV caches for both the starting and recent tokens. Consequently, in long-context scenarios, the memory usage and inference time for StreamingLLM still exhibit quadratic growth.

On the other hand, CEPE demonstrates both time and space efficiency by computing cross-attention solely between the input to the decoder (such as a user query) and the encoder. This approach allows CEPE to achieve subquadratic complexity concerning long contexts. However, it focuses on token-level embeddings instead of chunk-level embeddings, which necessitates more time and memory compared to E2LLM.

Furthermore, LongLLMLingua modifies the large language model (LLM) into a cross-encoder to identify the most relevant chunks and tokens related to the user query. Consequently, while its runtime increases dramatically with longer contexts due to the cross-encoder's high complexity, the memory usage remains stable. This is because the chunks can be processed sequentially, preventing significant memory overhead.

A similar trend is observed in another advanced prompt compression method, RAG. As we do not account for the memory costs associated with the retrieval process, and considering the retriever only recalls the 40 most relevant chunks from a lengthy context regardless of its total length, the generator's inference memory does not depend on context length. Nonetheless, since it processes the retrieved context token-by-token, the inference time and memory requirements still exceed those of E2LLM.

Lastly, LLoCO also enhances inference time through soft prompt compression; however, its text encoder, AutoCompressor, can only compress the original text by a maximum of 32 times, whereas E2LLM achieves an impressive compression factor of around 100 times. Furthermore, while Auto-Compressor processes all chunks sequentially, E2LLM leverages parallel processing, further minimizing inference time.

## H  RESULTS AND DISCUSSIONS ON ABLATION STUDIES

In this subsection, we conduct ablation studies of E2LLM using the QMSum and NarrativeQA datasets, which serve as representative benchmarks for long-context summarization and document question-answering tasks, respectively. Details of each variant examined in Table 9 are outlined below.

- $-$**Und** variant entails excluding the "understanding" task from our model and only employing the "reasoning" task for training purposes, which emphasis on the critical role that the "understanding" task plays within the model's performance.
- $-\mathcal{E}$ denotes the freezing of encoder parameters, thereby allowing only the adapter and the decoder-only LLM to be trainable. This configuration aims to substantiate our hypothesis that a pretrained encoder alone is incapable of preserving the pertinent information that significantly impacts the performance of the LLM. Hence, maintaining the encoder's parameters as trainable is crucial.
- $-\mathcal{D}$ entails keeping the decoder-only LLM frozen, in order to test whether the LLM can still adequately comprehend the output tokens from the adapter in the absence of any dedicated training.
- $+$**Overlap** variant introduces an overlap of 30% of the chunk size between sequential chunks during the chunking process. Moreover, within the scope of the "understanding" task's restatement operation, the model is required to restate the overlapping section of these chunks once.

Table 9: Ablation Study on QMSum and NarrativeQA.

| Variants | QMSum | | | | NarrativeQA | | | Avg. |
| | R1 | R2 | RL | G-mean | Prec. | Recall | F1 | Rel. Diff. |
|---|---|---|---|---|---|---|---|---|
| E2LLM | 25.37 | 6.55 | 18.75 | 14.61 | 13.53 | 13.79 | 12.35 | - |
| -Und | 22.64 | 4.86 | 16.20 | 12.13 | 11.13 | 10.04 | 9.94 | -16.39% |
| -$\mathcal{E}$ | 23.43 | 5.41 | 17.31 | 12.99 | 12.47 | 11.25 | 10.83 | -9.08% |
| -$\mathcal{D}$ | 23.09 | 4.93 | 17.11 | 12.49 | 12.23 | 10.95 | 10.46 | -12.03% |
| +Overlap | 25.23 | 6.39 | 17.95 | 14.25 | 13.28 | 13.94 | 12.41 | +1.78% |
| +BGE | 23.77 | 6.07 | 17.84 | 13.70 | 12.89 | 12.03 | 11.36 | -4.33% |
| +Llama2-13B | 25.77 | 6.72 | 18.89 | 14.84 | 13.75 | 13.74 | 12.68 | +4.70% |

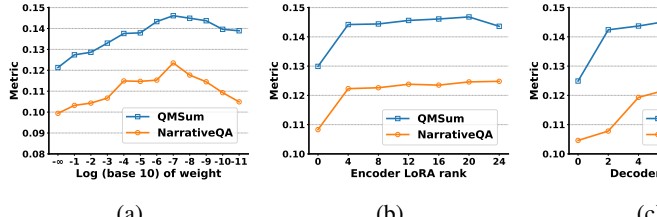

| (a) | (b) | (c) | (d) |

Figure 6: Effect of the hyperparameter. (a) the loss weight of "understanding" task. (b) the lora rank of encoder. (c) the lora rank of decoder. (d) the numer of layers in the adapter.

- +**BGE** variant test, on the other hand, involves replacing the GTE-Large-en model with the BGE-m3 model as the encoder. This study seeks to affirm that our model maintains compatable with different sentence-embedding models serving as encoders.

- +**Llama2−13B** configuration, similar in testing to the +BGE variant, is designed to verify the compatibility of our E2LLM with other LLMs serving as decoders.

First, we assess the significance of the "understanding" task within E2LLM. Our findings indicate a substantial decrease in performance—by 16.39%—when this task is omitted, highlighting its crucial role in helping E2LLM interpret the chunk embeddings produced by the encoder and further enhancing the performance of the "reasoning" task. Next, we examine the necessity of training the LoRA branches of the encoder and the decoder during alignment. As shown in Table 9, the results for configurations -$\mathcal{E}$ and -$\mathcal{D}$ underscore the importance of training these components; without this training, E2LLM's performance diminishes by 9.08% and 12.03%, respectively. Finally, we explore the impact of replacing the chunker, text encoder, and LLM decoder within E2LLM (notated as +overlap, +BGE, and +Llama2-13B). Our analysis reveals that chunkers with overlapping segments (e.g., 30% overlap) provide a modest performance boost. Additionally, employing more advanced encoders and decoders further enhances E2LLM's performance, suggesting that improvements in individual components can positively affect the overall system.

### H.1 SCALING TO LARGER-SCALE MODELS

We adopt Llama2-70B as the decoder to further validate the feasibility of E2LLM on larger-scale language models (denoted as E2LLM-70B). During training, we apply 4-bit quantization using QLoRA's Parameter-Efficient-Finetuning (PEFT) method. We conduct training and evaluating on QMSum, assessing its performance using the R1, R2, RL, G-mean, and PPL metrics. The results are shown in Table 10.

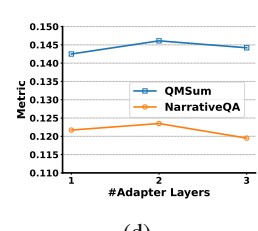

Table 10: Performance on E2LLM with larger-scale model.

| | R1 | R2 | RL | G-mean | PPL |
|---|---|---|---|---|---|
| E2LLM-7B | 0.2537 | 0.0655 | 0.1875 | 0.1461 | 13.68 |
| E2LLM-70B | 0.2561 | 0.0652 | 0.2312 | 0.1569 | 11.98 |
| Improv. | +0.95% | -0.458% | +21.99% | +7.39% | +12.43% |

Table 11: Effect of chunk size on the model performance.

| Chunk Size | Context Window | QMSum | | | | GovReport | | | | Quality | | | NarrativeQA | | | TriviaQA | | |
|---|---|---|---|---|---|---|---|---|---|---|---|---|---|---|---|---|---|---|
| | | R1 | R2 | RL | G-mean | R1 | R2 | RL | G-mean | Prec. | Recall | F1 | Prec. | Recall | F1 | Prec. | Recall | F1 |
| 128 | 100K | 24.29 | 6.35 | 18.81 | 14.26 | 29.98 | 9.29 | 17.21 | 16.86 | 12.94 | 14.76 | 12.54 | 13.42 | 13.65 | 12.11 | 32.95 | 33.90 | 32.89 |
| 512 | 400K | 25.37 | 6.55 | 18.75 | 14.61 | 33.14 | 10.75 | 18.59 | 18.78 | 13.44 | 14.95 | 12.94 | 13.53 | 13.79 | 12.35 | 33.22 | 34.51 | 33.37 |
| 1024 | 800K | 25.75 | 6.81 | 18.74 | 14.87 | 32.73 | 10.87 | 18.41 | 18.72 | 13.17 | 14.53 | 12.68 | 13.25 | 13.16 | 11.95 | 33.14 | 34.26 | 33.05 |
| 2048 | 1.6M | 24.13 | 6.33 | 18.01 | 14.01 | 30.12 | 9.03 | 17.04 | 16.67 | 12.56 | 14.03 | 12.17 | 13.07 | 12.93 | 11.74 | 32.07 | 31.94 | 31.36 |

As shown in the table, the performance of E2LLM significantly improves when using Llama2-70B, particularly in terms of Rouge-L and PPL. It is important to note that Rouge-1 and Rouge-2 evaluate unigram and bigram overlaps, respectively, measuring the match between the generated text and reference text at the word and phrase levels. In contrast, Rouge-L evaluates the similarity of the generated and reference texts based on the longest common subsequence (LCS), which measures structural similarity at the sentence level. This indicates that by leveraging a larger model, E2LLM is able to better capture the overall sentence structure and word order. Additionally, the reduction in PPL further demonstrates the model's ability to generate more coherent and reasonable content.

## H.2 Hyperparameter Sensitivity

In this section, we explore the effects of hyperparameters on the performance of E2LLM, specifically focusing on the weight assigned to the "understanding" task, the LoRA rank of the encoder and decoder, the number of layers in the adapter network, and the chunk size.

The weight assigned to the "understanding" task indicates its relative importance compared to the "reasoning" task. Recall that the input context typically has a much longer length than answers, making it too long to be fully reconstructed at once. To address this, we employ a sliding window approach, reconstructing the original context in segments based on a few consecutive chunks until the entire input has been reconstructed. Consequently, the samples for the "understanding" task are significantly more numerous than those for the "reasoning" tasks. To maintain sample balance, we usually assign a smaller weight to the restatement task. As depicted in Figure 6, the optimal weight may vary across different datasets, which may be influenced by factors such as context length and the sentence embedding model's capacity to comprehend the specific semantics of the context.

Moreover, we investigate the optimal LoRA rank of the encoder (i.e., GTE-Large-en) and the decoder (i.e., Llama2-7B-Chat) within the range of {0, 4, 8, 12, 16, 20, 24} and {0, 2, 4, 6, 8, 10, 12}, respectively. The findings suggest that having no trainable parameters—in other words, completely "freezing" the encoder and decoder—hinders the effective extraction of original context content and alignment between the encoder and decoder, as discussed in Section 3.1. As the rank of the two modules increases, a corresponding improvement in performance is observed, thereby underscoring the importance of training. Performance enhancement continues until it reaches a peak within a specific range of ranks. However, beyond this optimal range, further increases in rank lead to a decline in performance, attributable to overfitting on the training datasets.

We also examine the impact of the number of layers in the adapter network. Figure 6 shows that a two-layer MLP consistently delivers superior performance across different datasets, indicating stability in results. We hypothesize that a single-layer MLP may struggle with the alignment task, while a three-layer MLP might lead to overfitting on the training data.

We investigate the effect of chunk size on model performance, experimenting with sizes of 128, 512, 1024, 2048 characters, corresponding to maximum context window sizes of 100K, 400K, 800K, and 1.6M tokens for various E2LLM variants. Results in Table 11 show that the differences in performance metrics across different chunk sizes are relatively small for all datasets used in this study, indicating that the alignment process in E2LLM can effectively mitigate the impact of chunk size on performance. Nonetheless, selecting an optimal chunk size can still provide a slight performance boost. While smaller chunks might reduce compression and better preserve inputs, they may hinder context capture in longer sentences or paragraphs, making it difficult for the encoder to grasp semantics, which affects downstream tasks. Conversely, larger chunk sizes increase diversity and

noise, complicating semantic capture and leading to decreased performance, especially in tasks like DocumentQA where relevant sentences may be overlooked.

# I  DISCUSSION AND OUTLOOK

## I.1  CONTINUE PRETAINING USING THE E2LLM FRAMEWORK

While E2LLM demonstrates comparable or superior performance compared to various baseline methods when utilizing the same amount of fine-tuning data, it has become increasingly common to engage in continue-pretraining (CPT) alongside supervised fine-tuning (SFT). This approach typically involves leveraging substantial quantities of high-quality, long-context data. By doing so, systems can achieve a long-text language model (LLM) that exhibits versatility across diverse long-text tasks. This practice has been effectively illustrated by recent models in the Llama and Qwen series (Dubey et al., 2024; Yang et al., 2024; Hui et al., 2024).

In this context, we will explore the methodology for conducting CPT within the E2LLM framework. Given that E2LLM functions akin to an encoder-decoder architecture, it is logical to adopt pretraining tasks prevalent in other established encoder-decoder frameworks, such as T5 (Raffel et al., 2019), BART (Lewis et al., 2020), and GLM (Du et al., 2022). However, considering the unique characteristics of the E2LLM model, we specifically recommend the use of prefix language modeling (PLM) (Wang et al., 2022) as the pretraining task. This choice ensures that the CPT process aligns seamlessly with the subsequent SFT process.

Concretely, suppose that the chunk size of the text encoder is $C$ and the length of the decoder is $L$. In this setup, we can create random pretraining sequences with lengths of $\ell C + L - \ell$, where $1 \leq \ell \leq L - 1$ represents the number of chunks. We then can partition the prefix segments of length $\ell C$ into $\ell$ individual chunks, which are then fed into the text encoder. The task for E2LLM during this CPT phase is to predict the remaining segments of length $L - \ell$. This structured approach not only enhances the model's ability to comprehend and generate lengthy texts but also sets a solid foundation for effective fine-tuning on targeted applications thereafter.

## I.2  LEARNABLE CHUNK SIZE

While the performance of E2LLM is not highly sensitive to chunk size, selecting the optimal size can enhance its effectiveness, as illustrated in Table 11. This raises an intriguing question: Can we determine a chunk size that further boosts the performance of E2LLM? We believe there are two primary approaches to achieving this goal.

First, we can apply techniques commonly used to optimize hyperparameters in neural networks or during neural architecture search (NAS) to the chunk size learning process. Approaches such as Bayesian optimization, reinforcement learning, and meta-learning can be adapted to optimize both the chunk size and E2LLM model parameters simultaneously.

Second, chunk size can also be optimized independently of the E2LLM model parameters. One promising strategy is to explore chunking based on the semantic relationships between tokens, such as through meta-chunking (Zhao et al., 2024). However, a significant challenge with existing chunking methods is that the tokens within a chunk only capture information from that specific chunk, resulting in a loss of contextual information from nearby chunks. To address this issue, we can consider the "late" chunking method (Günther et al., 2024). This approach first embeds all tokens of a long text using a long-context embedding model and then applies chunking by mean pooling the token embeddings within each chunk. While this method provides chunk embeddings that encapsulate full contextual information, it comes with the drawback of increased complexity, as the computational demands of text encoders scale quadratically with the length of the input text.

