# OpenReview forum: "E2LLM: Encoder Elongated Large Language Models for Long-Context Understanding and Reasoning"
_ICLR.cc/2025/Conference — Submitted to ICLR 2025_

### Official Review · Reviewer_uKEN · 2024-10-20

**Soundness:** 3
**Presentation:** 3
**Contribution:** 2
**Rating:** 5
**Confidence:** 3

**Summary:**

This paper presents E2LLM, a novel approach to tackle the challenge of processing long contexts in LLMs. The authors identify three key challenges: performance (T1), efficiency (T2), and compatibility (T3). E2LLM addresses these challenges by (1) splitting long contexts into chunks; (2) compressing each chunk into an embedding vector using a pre-trained text encoder; (3) utilizing an adapter to align the encoder’s output with the decoder’s input embedding space; (4) training two objectives: reconstructing the original text from the encoder’s output and fine-tuning the LLM on long-context instruction tasks.

**Strengths:**

1. E2LLM achieves a good balance between performance, efficiency, and compatibility;

2. The model is trained end-to-end, ensuring the alignment between the encoder and decoder and minimizing information loss;

3. Extensive experiments demonstrate that E2LLM outperforms several existing methods on long-context tasks such as document summarization, QA, Needle-in-a-Haystack, as well as LongBench;

4.  E2LLM achieves the lowest inference time and memory usage among compared methods.

**Weaknesses:**

1. The authors seem to overclaim that "E2LLM solves all the challenges of the impossible triangle" since the performance on long-context tasks of E2LLM still falls far behind "continual pertaining on longer texts + long-context SFT" paradigm [1, 2] (though there is no comparison between E2LLM and this approach in the paper), which is the mainstream and SoTA approach now and widely used in most advanced LLMs such as Llama3.1 and GLM4. This makes the real-world applicability of E2LLM doubtful. A comparison between E2LLM and "continual pretraining + SFT" approach should be presented to support the claim.

2. The figure of Needle-in-a-Haystack is deceptive since the color is all green (the score less than 5 should be marked in red). Though outperforms the baseline methods, E2LLM still struggles in recalling the needle with only 30% success rate， while current long-context LLMs can achieve nearly 100% performance.

[1] Effective Long-Context Scaling of Foundation Models
[2] LongAlign: A Recipe for Long Context Alignment of Large Language Models

**Questions:**

1. What is the maximum compression rate of E2LLM such that the context can be recovered losslessly?

2. Can E2LLM effectively harness the longer context windows? In other words, does the PPL/performance of E2LLM continue decreasing/increasing when the context window becomes longer?

---

> ### Author Response · Authors · 2024-11-28
> **Response to Reviewer uKEN (Part 1)**
>
> We are deeply grateful for the reviewer’s valuable comments, which helps enhance the quality of our work. We have crafted detailed responses for each suggestion and incorporated them into the revised manuscript. In our response, comments from the reviewer are highlighted in italics, with our responses directly following. For clarity, quotations from the adjusted manuscript are presented in markdown quotation mode. The corresponding modifications in the paper are highlighted in blue. Unless otherwise stated in our response, all pages, equations, sections, and bibliographical references refer to those in the revised paper.
>
>
> _W1 - The authors seem to overclaim that "E2LLM solves all the challenges of the impossible triangle" since the performance on long-context tasks of E2LLM still falls far behind "continual pertaining on longer texts + long-context SFT" paradigm [1, 2] (though there is no comparison between E2LLM and this approach in the paper), which is the mainstream and SoTA approach now and widely used in most advanced LLMs such as Llama3.1 and GLM4. This makes the real-world applicability of E2LLM doubtful. A comparison between E2LLM and "continual pretraining + SFT" approach should be presented to support the claim._
>
> We would like to argue that **most long-context extension methods aim to enhance the context window with minimal to no fine-tuning**, as explicitly stated in studies like Position Interpolation (Chen et al. (2023a)) and YaRN (Peng et al., 2023). This is consistent with E2LLM’s goal of compatibility with pretrained models. As discussed in Section 1 (Lines 041-044),
>
> > Achieving this compatibility is crucial for effectively leveraging the pretrained knowledge contained in these models, allowing for **parameter and sample efficiency **without necessitating extensive additional training with large datasets.**
>
> This rationale underpins our decision not to focus on continue-pretraining (CPT) plus supervivsed fine tuning (SFT) in the first place, as it **deviates from the original motivation behind long-context extension methods**. Our core aim is to demonstrate that, given an equivalent training dataset, E2LLM—being tuned with fewer parameters—can outperform baseline methods that necessitate more extensive tuning. This assertion is validated in Section 4 (see Tables 1-2 and Figure 3), where E2LLM effectively harnesses the knowledge embedded in the pretrained text encoder and LLM decoder.
>
> On the other hand, **we acknowledge that CPT plus SFT is a prevalent practice to achieve practical long-context LLMs, as seen in the Llama and Qwen series**. However, due to resource constraints and time limitations, we could not conduct CPT plus SFT using the E2LLM framework on a substantial amount of high-quality long-context data. Instead, **we provide a discussion in Appendix I.1 (Lines 1303-1323 on Page 25) regarding how CPT can be conducted within the E2LLM framework** as follows:
>
> > While E2LLM demonstrates comparable or superior performance compared to various baseline methods when utilizing the same amount of fine-tuning data, it has become increasingly common to engage in continue-pretraining (CPT) alongside supervised fine-tuning (SFT). This approach typically involves leveraging substantial quantities of high-quality, long-context data. By doing so, systems can achieve a long-text language model (LLM) that exhibits versatility across diverse long-text tasks. This practice has been effectively illustrated by recent models in the Llama and Qwen series (Dubey et al., 2024; Yang et al., 2024; Hui et al., 2024).
>
> > In this context, we will explore the methodology for conducting CPT within the E2LLM framework. Given that E2LLM functions akin to an encoder-decoder architecture, it is logical to adopt pretraining tasks prevalent in other established encoder-decoder frameworks, such as T5 (Raffel et al., 2019), BART (Lewis et al., 2020), and GLM (Du et al., 2022). However, **considering the unique characteristics of the E2LLM model, we specifically recommend the use of prefix language modeling (PLM) (Wang et al., 2022) as the pretraining task.** This choice ensures that the CPT process aligns seamlessly with the subsequent SFT process.
>
> > Concretely, suppose that the chunk size of the text encoder is $C$ and the length of the decoder is $L$. In this setup, we can create random pretraining sequences with lengths of $\ell C + L - \ell$, where $1 \leq \ell \leq L-1$ represents the number of chunks. We then can partition the prefix segments of length $\ell C$ into $\ell$ individual chunks, which are then fed into the text encoder. The task for E2LLM during this CPT phase is to predict the remaining segments of length $L - \ell$. This structured approach not only enhances the model's ability to comprehend and generate lengthy texts but also sets a solid foundation for effective fine-tuning on targeted applications thereafter.

---

> ### Author Response · Authors · 2024-11-28
> **Response to Reviewer uKEN (Part 2)**
>
> _W2 - The figure of Needle-in-a-Haystack is deceptive since the color is all green (the score less than 5 should be marked in red). Though outperforms the baseline methods, E2LLM still struggles in recalling the needle with only 30% success rate， while current long-context LLMs can achieve nearly 100% performance._
>
>
> In the original version, we use a green-toned color bar to represent accuracy levels, with varying shades indicating different accuracy values. **Following your suggestion, we have updated the color bar to a green-to-red gradient, which enhances the intuitiveness of the visualization.** The revised figure can be found in Figure 3 on Page 9.
>
>
>
> Regarding the accuracy on the Needle In A Haystack (NIAH) task, we believe it is influenced by several factors, such as whether large-scale continual pretraining has been conducted and the parameter size of the model. Both factors directly impact the performance of LLMs on NIAH tasks. In our experiments, neither our method nor the baselines undergo large-scale continual pretraining. Indeed, **we only used 13K samples to train all models**. Furthermore, we use the relatively small and limited LLaMA2-7B model as the backbone, which inherently restricts the accuracy achievable, making 100% unattainable in this context.
>
>
>
> To clarify this point, we have mentioned the number of samples we used to train all models in Appendix D (Lines 1006-1012) as:
>
> > Note that we use all training samples from QMSum, GovReport, Quality, and NarrativeQA, but randomly select 3,000 samples from TriviaQA, as the sample size of this dataset is much larger than that of others, but the average sequence length is the shortest. **The total number of training samples is around 13,000.**

---

> ### Author Response · Authors · 2024-11-28
> **Response to Reviewer uKEN (Part 3)**
>
> _Q1 - What is the maximum compression rate of E2LLM such that the context can be recovered losslessly?_
>
> First, we would like to clarify that the reconstruction or the "understanding" task is an auxiliary task in E2LLM. Our main focus is on the "reasoning" task such as document summarization and QA. To make this distinction clear, we have elaborated on it in Section 3.2 (Lines 264-269, Page 5):
>
> > It is important to note that **the “understanding” task serves as an auxiliary task, while our primary focus remains on the “reasoning” task**. We determine the final checkpoints exclusively based on the validation loss associated with the “reasoning” task. In this context, we do not anticipate that E2LLM can achieve lossless compression of the context. However, we believe that the LLM decoder is capable of retaining or comprehending essential information from the context. The LLM operates as a "suggestion feature" for input methods, leveraging hints to generate meaningful responses. In this case, the chunk tokens provided by the text encoder serve as these essential hints.**
>
> On the other hand, we acknowledge that it is interesting to check whether E2LLM can stably recover the context. To this end, we conduct an experiment using QMsum training data with chunk sizes of 128, 512, 1024, and 2048. We evaluated the Precision, Recall, and F1-score of the reconstructed sequences across various lengths (100, 500, 1000, 2000, 3000 tokens) in relation to the ground truth. The results of this evaluation are summarized  in the following Table.
>
> | Chunk size | 0.1K  |       |       | 0.5K  |       |       | 1K    |       |       | 2K    |      |       | 3K    |      |       |
> | ---------- | ----- | ----- | ----- | ----- | ----- | ----- | ----- | ----- | ----- | ----- | ---- | ----- | ----- | ---- | ----- |
> | Metric     | Prec. | Rec.  | F1    | Prec. | Rec.  | F1    | Prec. | Rec.  | F1    | Prec. | Rec. | F1    | Prec. | Rec. | F1    |
> | 128        | 7.04  | 11.53 | 7.83  | 11.39 | 11.34 | 10.63 | 26.27 | 6.30  | 8.78  | 49.42 | 4.74 | 8.00  | 56.41 | 3.18 | 4.31  |
> | 512        | 13.95 | 19.11 | 14.78 | 20.59 | 14.97 | 16.88 | 28.54 | 13.51 | 16.34 | 44.26 | 8.63 | 13.45 | 47.48 | 6.36 | 10.04 |
> | 1024       | 12.86 | 17.30 | 12.03 | 18.39 | 14.65 | 15.97 | 27.44 | 13.64 | 17.07 | 45.16 | 7.99 | 12.08 | 54.46 | 5.92 | 8.97  |
> | 2048       | 5.83  | 12.72 | 7.16  | 18.23 | 11.82 | 11.60 | 24.78 | 10.65 | 13.11 | 40.18 | 6.50 | 11.44 | 45.00 | 4.37 | 7.79  |
>
>
> Our findings indicate that Precision consistently increases with the length of the reconstructed sequence, while Recall gradually declines, irrespective of the chunk size. The F1 score demonstrates an initial increase with longer sequence lengths, followed by a decline. This trend can be attributed to the fact that the reconstruction task is generally oriented toward reconstructing sequences of approximately 3000 characters (or 500-600 tokens) during training. During inference, if the length of the true sequence is shorter than 500 tokens, the generated sequence tends to be longer than the ground truth. Conversely, when the true sequence exceeds 600 tokens, the generated output tends to be shorter than the ground truth. As a result, the generated sequences often introduce additional tokens when the true sequence is shorter than 500 tokens, leading to lower Precision but relatively higher Recall. In contrast, when the true sequence is longer than 600 tokens, the model struggles to capture all tokens from the ground truth, resulting in higher Precision but lower Recall.
>
> Moreover, we can find that optimal results are achieved with a chunk size of 1024, which aligns with the observations in our ablation study regarding chunk size (Appendix H.2, Page 24-25, Lines 1288-1297).

---

> ### Author Response · Authors · 2024-11-28
> **Response to Reviewer uKEN (Part 4)**
>
> _Q2 - Can E2LLM effectively harness the longer context windows? In other words, does the PPL/performance of E2LLM continue decreasing/increasing when the context window becomes longer?_
>
>
> Thank you for your valuable comment.
>
> In response, we have investigated the sensitivity of E2LLM's performance to variations in context length for the QMsum and NarrativeQA datasets in Appendix C (Page 18, Lines 956-960) as follows:
>
> > Next, we investigate the sensitivity of the models' performance to variations in context length. To do this, we categorize samples from the QMsum and NarrativeQA datasets into five groups based on their context lengths and then evaluate the perplexity (PPL) of the answers within each group. Our findings are summarized in Table 6.
>
> Table 6-1: Performance as a function of context length. The best results are in bold, the second are underlined, and the third are wavy underlined.
> | Methods       | 0K-6K   |  | 6K-12K  |  | 12K-18K |  | 18K-24K|   | 24K+    |  |
> | ------------- | --------- | --------- | --------- | --------- | --------- | --------- | --------- | --------- | --------- | --------- |
> || G-mean        | PPL       | G-mean    | PPL       | G-mean    | PPL       | G-mean    | PPL       | G-mean    | PPL       |
> | Llama2-7B     | 13.05     | 28.57     | 11.99     | 85.35     | 11.54     | 84.31     | 12.56     | 81.74     | 10.32     | 85.60     |
> | StreamingLLM  | 3.27      | 36.35     | 4.21      | 168.63    | 3.32      | 224.24    | 3.26      | 356.17    | 2.45      | 362.41    |
> | LongLoRA      | 5.91      | 12.92     | 8.13      | 13.17 | 8.3       | 14.65     | 9.66      | 15.97     | 7.44      | 17.31     |
> | CEPE          | 11.66     | 128.01    | 10.42     | 144.34    | 9.29      | 161.28    | 8.21      | 145.54    | 6.56      | 234.24    |
> | YaRN          | 13.57     | 14.52     | 12.10     | 14.02     | 12.88     | 17.06     | 11.49     | 17.75     | 6.33      | 18.90     |
> | RAG           | 6.12      | 17.94     | 8.72      | 17.58     | 9.65      | 20.95     | 9.03      | 19.59     | 6.24      | 19.39     |
> | LongLLMLingua | 7.73      | 11.25 | 9.83      | 15.12     | 8.72      | 16.25     | 9.08      | 19.66     | 8.87      | 21.55     |
> | LLoCO         | 13.63     | 34.56     | 12.78     | 41.27     | 13.15     | 47.45     | 12.13     | 47.87     | 10.03     | 56.30     |
> | E2LLM         | 15.04 | 12.69     | 15.27 | 13.47     | 14.14 | 13.95 | 14.26 | 13.33 | 15.31 | 13.92 |
>
> >
>
> Table 6-2: Performance as a function of context length. The best results are in bold, the second are underlined, and the third are wavy underlined.
> | Methods       | 0K-24K |   | 24K-48K |  | 48K-72K |  | 72K-96K |  | 96K+  |  |
> | ------------- | --------- | --------- | --------- | --------- | --------- | --------- | --------- | --------- | --------- | --------- |
> || F1            | PPL       | F1        | PPL       | F1        | PPL       | F1        | PPL       | F1        | PPL       |           |
> | Llama2-7B     | 3.10      | 75.81     | 10.71     | 178.28    | 7.51      | 250.81    | 0.61      | 2303.08   | 2.48      | 2215.08   |
> | StreamingLLM  | 4.36      | 79.34     | 2.53      | 135.71    | OOM       | OOM       | OOM       | OOM       | OOM       | OOM       |
> | LongLoRA      | 3.23      | 11.93 | 9.47      | 12.17 | OOM       | OOM       | OOM       | OOM       | OOM       | OOM       |
> | CEPE          | 3.37      | 3568.12   | 2.65      | 2272.04   | OOM       | OOM       | OOM       | OOM       | OOM       | OOM       |
> | YaRN          | 7.19      | 13.94     | 6.59      | 17.16     | OOM       | OOM       | OOM       | OOM       | OOM       | OOM       |
> | RAG           | 2.40      | 12.98     | 2.14      | 41.35     | 2.55      | 60.28     | 2.14      | 58.32     | 1.43      | 57.20     |
> | LongLLMLingua | 7.84      | 26.52     | 6.23      | 29.45     | 3.16      | 29.96     | 1.72      | 38.53     | 1.03      | 48.53     |
> | LLoCO         | 10.89     | 13.32     | 10.67     | 15.67     | 10.88     | 17.31     | 11.42     | 16.19     | 9.43      | 18.54     |
> | E2LLM         | 12.12 | 13.45     | 12.41 | 12.87     | 12.76 | 12.96 | 12.23 | 13.65 | 11.97 | 13.71 |
>
>
> > The results presented in the table indicate that **E2LLM demonstrates a strong resilience to variations in context length for both summarization (QMsum) and question-answering (NarrativeQA) tasks, typically achieving the best results among all models.** This robustness can be attributed to the "understanding" task incorporated during the training of E2LLM (see Section 3.2). By reconstructing different parts of the context, E2LLM effectively comprehends the information, regardless of its length.

---

> ### Author Response · Authors · 2024-11-28
> **Response to Reviewer uKEN (Part 5)**
>
> > Notably, the performances of YaRN, LongLoRA, CEPE, RAG, and LongLLMLingua also exhibit insensitivity to context length. On the other hand, LLoCO's performance declines slowly with increasing context length. Finally, streamingLLM and the original Llama2-7B demonstrate sensitivity to context length; streamingLLM loses more information in the middle of the context as length increases due to its specific $\Lambda$-shaped attention mask, while Llama2-7B struggles to handle long contexts altogether, as its maximum length has not been extended.

---

> ### Author Response · Authors · 2024-12-02
> **Seeking for your further feedback**
>
> Dear Reviewer uKEN,
>
>
> We would like to thank you once again for your constructive feedback on our paper. We have made significant efforts to address the concerns you raised and have revised the manuscript accordingly. Given that the discussion period is nearing its end, we kindly ask if you could review the updated version and let us know if there are any remaining issues or points that require further clarification.
>
> We are more than happy to address any additional questions or concerns you might have.
>
> Thank you for your time and consideration.

---

### Official Review · Reviewer_JbEJ · 2024-10-31

**Soundness:** 2
**Presentation:** 3
**Contribution:** 3
**Rating:** 6
**Confidence:** 3

**Summary:**

This paper introduces E2LLM, designed to efficiently and effectively handle long-context understanding in LLMs. The method balances long-context performance, computational efficiency, and compatibility. E2LLM employs a soft prompt compression technique, segmenting long contexts into chunks, compressing them into embeddings using a pretrained text encoder, and aligning these embeddings with a decoder-only LLM through an adapter. The proposed training objectives of the model focus on reconstructing encoder outputs and fine-tuning for long-context instructions. Experiments demonstrate that E2LLM outperforms existing SoTA methods across multiple long-context benchmarks including summarization, QA, needle-in-a-haystack, and LongBench, while achieving low inference time and memory usage.

**Strengths:**

1. The paper is overall well-written and easy to follow.
2. The integration of an encoder to compress long contexts while preserving essential information is innovative and leverages existing pretrained architectures efficiently.
3. The extensive experiments show the proposed method's performance, efficiency, and robustness. The method could serve as a strong baseline for future works.

**Weaknesses:**

1. The authors would like to introduce E2LLM as a general framework that could work with any LLMs, which is intended to show the method's flexibility. However, when compared to similar methods like CEPE, both the encoder and the LoRA rank used are different. This both makes the "Trainable Parameters" in Table 1 incomparable and makes me doubt whether the improvement comes from the more powerful GTE-Large-en encoder or the proposed method.
2. It is unsure how the authors trained E2LLM and the baselines on the tasks specified in Appendix C. How were the hyperparameters for each method decided?
3. The analysis of E2LLM's complexity has flaws. As per Figure 2, the input contains a span (prompt + query) that is directly put into the decoder. The complexity needs to separate the computation of the "context" part and the "prompt" part.
4. The paper misses some related works. For example, for position embedding extension methods, there have been better methods [1] or improvements to YaRN [2]. For methods similar to CEPE, the concurrent work FocusLLM [3] has a very similar design to E2LLM.

[1] Guanzheng Chen, Xin Li, Zaiqiao Meng, Shangsong Liang, and Lidong Bing. 2023. CLEX: Continuous Length Extrapolation for Large Language Models. In ICLR 2024.

[2] Suyuchen Wang, Ivan Kobyzev, Peng Lu, Mehdi Rezagholizadeh, and Bang Liu. 2024. Resonance RoPE: Improving Context Length Generalization of Large Language Models. In ACL 2024 Findings.

[3] Zhenyu Li, Yike Zhang, Tengyu Pan, Yutao Sun, Zhichao Duan, Junjie Fang, Rong Han, Zixuan Wang, and Jianyong Wang. 2024. FocusLLM: Scaling LLM’s Context by Parallel Decoding. arXiv Preprint.

**Questions:**

Other than the problems mentioned in the weaknesses section, below are some other questions:
1. In line 396, you mentioned "E2LLM addresses these challenges...", which includes the challenge that StreamingLLM requires training. However, E2LLM also requires training to align the encoder, adaptor, and decoder. Please explain this claim.
2. In line 404, you mentioned that YaRN faces the challenge that "attention mechanisms can become dispersed in exceedingly long contexts". However, YaRN proposes the dynamic scaling method for the attention scores to alleviate this problem.
3. In line 265, it would be easier to understand if "sequence length" could be replaced by "context window" for the input capacity of the model.

---

> ### Author Response · Authors · 2024-11-28
> **Response to Reviewer JbEJ (Part 1)**
>
> We greatly value the reviewer’s constructive feedback, which has been instrumental in refining both the presentation and the substance of our paper. In response, we have carefully addressed each comment and provided a detailed explanation to ensure all concerns are thoroughly resolved. In our response, comments from the reviewer are highlighted in italics, with our responses directly following. For clarity, quotations from the adjusted manuscript are presented in markdown quotation mode. The corresponding modifications in the paper are **highlighted in blue**. Unless otherwise stated in our response, all pages, equations, sections, and bibliographical references refer to those in the revised paper.
>
>
>
> _W1 - The authors would like to introduce E2LLM as a general framework that could work with any LLMs, which is intended to show the method's flexibility. However, when compared to similar methods like CEPE, both the encoder and the LoRA rank used are different. This both makes the "Trainable Parameters" in Table 1 incomparable and makes me doubt whether the improvement comes from the more powerful GTE-Large-en encoder or the proposed method._
>
>
> First, we would like to emphasize the difference in the number of trainable parameters as illustrated in Table 1. CEPE has a significantly higher number of trainable parameters (1.31B) compared to E2LLM (16M). Despite this disparity, E2LLM demonstrates superior performance, highlighting its effectiveness even with fewer parameters. We believe this comparison is valid and underscores the strength of E2LLM.
>
>
>
> Second, while the GTE-Large-en encoder is indeed a powerful component of E2LLM, one of the key advantages of our framework lies in its ability to fully leverage existing pretrained text encoders and LLM decoders. As indicated in Section 1 (Lines 041-043), a primary goal of E2LLM is to maintain compatibility with pretrained models (T3).
>
> > Achieving this compatibility is crucial for effectively leveraging the pretrained knowledge contained in these models, allowing for **parameter and sample efficiency** without necessitating extensive additional training with large datasets.
> >
>
> In contrast, **CEPE is constrained by its requirement for the encoder to align with the decoder's tokenizer, which limits its ability to directly benefit from pretrained models and necessitates extensive training of the encoder from scratch.**
>
>
>
> Furthermore, we would like to highlight the performance comparison with another baseline, RAG, which employs the same chunking strategy, the same GTE encoder, and the same LLM decoder as E2LLM. Despite these similarities, our proposed method consistently outperforms RAG, further demonstrating the unique strengths of E2LLM.
>
>
>
> In summary, **we attribute the success of E2LLM to its flexibility in seamlessly integrating powerful pretrained text encoders and LLM decoders, rather than relying exclusively on LLM decoders.**
>
> _W2 - It is unsure how the authors trained E2LLM and the baselines on the tasks specified in Appendix C. How were the hyperparameters for each method decided?_
>
>
> Thank you for pointing this out. **The parameter settings for E2LLM and the baselines are detailed in Appendix B on Page 15-16, i.e., Overview of Baseline Methods.** However, we realize that we have omitted the implementation details of E2LLM's adapter.
>
> We have now included the following description in Appendix B (Page 16-17, Lines 861-863, 874-875):
>
> _W3 - The analysis of E2LLM's complexity has flaws. As per Figure 2, the input contains a span (prompt + query) that is directly put into the decoder. The complexity needs to separate the computation of the "context" part and the "prompt" part._
>
>
> Thanks for pointing this out! We appreciate your observation regarding the contribution of the prompt to the complexity calculation.
>
>
>
> In response, we have clarified our discussion in Section 3.2 (Page 6 Lines 279-280) as:
>
> > Let us denote the original **context length (excluding the prompt or instruction)** as $L$ and the chunk size in E2LLM as $C$.
>
>
> Specifically, we have replaced "input length" by "context length (excluding the prompt or instruction)".
> > Regarding the Adapter, its structure is designed as a two-layer Multilayer Perceptron (MLP). The first layer's input and output neuron numbers correspond to the embedding dimensions of the encoder and decoder, respectively, with GELU used as the activation function. The second layer maintains equal input and output dimensions, aligning with the decoder's embedding size.

---

> ### Author Response · Authors · 2024-11-28
> **Response to Reviewer JbEJ (Part 2)**
>
> _W4 - The paper misses some related works. For example, for position embedding extension methods, there have been better methods [1] or improvements to YaRN [2]. For methods similar to CEPE, the concurrent work FocusLLM [3] has a very similar design to E2LLM._
>
>
> Thank you for pointing this out!
>
> We have cited and discussed CLEX and FocusLLM in our paper in Section 2 (Lines 124-126 and Lines 178-183) as:
>
> > Another approach, CLEX (Chen et al., 2024a), replaces manual design with learned scaling factors through neural differential equations, effectively overcoming the limitations inherent in traditional positional extrapolation techniques.
>
>
> > Additionally, we note that, concurrently with our work, FocusLLM (Li et al., 2024b) has also adopted a strategy of chunking long contexts and summarizing each chunk using the hidden states of the local context from all layers of an LLM. These hidden states are concatenated to serve as the key-value cache for the same LLM, providing answers to user queries. From the perspective of E2LLM, FocusLLM essentially employs an LLM as a text encoder, which influences both training and inference efficiency.
>
>
>
> Furthermore, **we have conducted a comparison of CLEX with E2LLM and other baseline methods.** Due to time constraints, we focused our evaluation on the QMsum dataset, following the settings outlined in Section 4.1. The results are summarized in the table below. While CLEX achieves a lower perplexity (PPL), its G-mean performance is inferior to that of E2LLM.
>
> Table R1: Performance of CLEX on QMSum.
>
> | Methods | QMSum | |
> | --- | :---: | --- |
> | | G-mean | PPL |
> | Llama2-7B | 11.52 | 84.92 |
> | StreamingLLM | 3.62 | 220.12 |
> | LM-Infinite | 4.73 | 180.43 |
> | LongLoRA | 8.98 | 14.48 |
> | CEPE | 10.77 | 154.16 |
> | YaRN | 12.31 | 16.22 |
> | RAG | 7.24 | 19.11 |
> | LongLLMLingua | 8.93 | 17.55 |
> | LLoCO | 12.99 | 46.32 |
> | CLEX | 14.42 | **10.99** |
> | E2LLM | **14.61** | 13.68 |
>
> _Q1 - In line 396, you mentioned "E2LLM addresses these challenges...", which includes the challenge that StreamingLLM requires training. However, E2LLM also requires training to align the encoder, adaptor, and decoder. Please explain this claim._
>
>
> E2LLM is designed to navigate the complexities of the "impossible triangle," which encompasses high performance, low complexity, and compatibility with pretrained models. While it is true that E2LLM necessitates some level of training to align the encoder, adaptor, and decoder, it is important to emphasize that this training is efficient.
>
>
>
> E2LLM is compatible with pretrained text encoders and LLM decoders, allowing us to achieve **parameter and sample efficiency without the need for extensive training on large datasets** (as noted in Lines 42-43 on Page 1). This efficiency is further supported by the results presented in Table 1 on Page 7, which demonstrate that **E2LLM typically achieves superior performance with a reduced number of trainable parameters when trained on the same amount of data.**
>
>
>
> In contrast, while StreamingLLM does not require training, its performance is generally inferior to that of other methods. Thus, it does not effectively address the challenges posed by the "impossible triangle."
>
> Q2 - _In line 404, you mentioned that YaRN faces the challenge that "attention mechanisms can become dispersed in exceedingly long contexts". However, YaRN proposes the dynamic scaling method for the attention scores to alleviate this problem._
>
>
>
> We would like to clarify the distinctions between the issues addressed by YaRN and the concept of attention dispersion.
>
>
>
> The dynamic scaling method only alleviates abrupt performance degradation when the context length exceeds what was seen during training, and **the primary focus of Yarn is to ensure that the position embeddings of nearby tokens remain distinguishable. This approach helps maintain the model's sensitivity to local context.**
>
>
>
> Conversely, **attention dispersion refers to the phenomenon where, as the number of input tokens increases, attention scores for each query become widely dispersed across a larger set of keys.** This can result in overlapping keys with different semantic meanings, complicating the model's ability to concentrate on relevant information.
>
>
>
> It is important to note that **YaRN does not directly address the problem of attention dispersion. Simply distinguishing position embeddings of tokens is not sufficient to resolve this issue, as position embeddings alone do not fully determine the attention scores.**
>
>
>
> A more straightforward approach for mitigating attention distraction is to reduce the number of input tokens, as pointed out in [R1]. Indeed, **E2LLM effectively decreases the number of input tokens through soft prompt compression.**
>
>
>
> [R1] Deng et al, FltLM: An Intergrated Long-Context Large Language Model for Effective Context Filtering and Understanding, 2024.

---

> ### Author Response · Authors · 2024-11-28
> **Response to Reviewer JbEJ (Part 3)**
>
> _Q3 - In line 265, it would be easier to understand if "sequence length" could be replaced by "context window" for the input capacity of the model._
>
>
> Thank you for your valuable feedback. Following your suggestion, we have replaced "sequence length" and "input length" with "context window" to enhance readability.

---

> ### Author Response · Authors · 2024-12-02
> **Seeking for your further feedback**
>
> Dear Reviewer JbEJ,
>
>
> We would like to thank you once again for your constructive feedback on our paper. We have made significant efforts to address the concerns you raised and have revised the manuscript accordingly. Given that the discussion period is nearing its end, we kindly ask if you could review the updated version and let us know if there are any remaining issues or points that require further clarification.
>
> We are more than happy to address any additional questions or concerns you might have.
>
> Thank you for your time and consideration.

---

> > ### Comment · Reviewer_JbEJ · 2024-12-02
> >
> > Thank you for the detailed clarifications. They address most of my concerns. However, regarding Weakness 1, while I appreciate the emphasis on compatibility as a strength of the proposed method, I still have reservations about the claim of better performance. Specifically, the differences in the encoder used and the LoRA rank compared to CEPE leave me uncertain about the extent to which the performance improvement stems from the proposed method itself rather than the more powerful GTE-Large-en encoder or a different LoRA rank. As such, I will maintain my original score.

---

> ### Author Response · Authors · 2024-12-02
> **Further Clarifications on Reviewer JbEJ‘s Feedback**
>
> Thank you for your feedback. We would like to further clarify the number of trainable parameters in CEPE and the encoder it uses in the experiment.
>
> Regarding CEPE, we would like to argue that **the composition of its trainable parameters, which includes the encoder that needs to align with the decoder’s tokenizer, as well as the additional randomly initialized cross-attention layers in each layer of the decoder’s transformer, results in its large number of parameters. Therefore, the number of CEPE's trainable parameters is fixed, and is not related to the LoRA rank**, which represent its inherent limitation on parameter efficiency.
>
> Moreover, as we previously mentioned, **due to the need to ensure consistency in vocabulary between the encoder and the decoder**, **CEPE** pre-trains a masked language model (MLM) on the RedPajama dataset. Therefore, it **cannot directly apply the pre-trained GTE-Large-en as the encoder**, which is another compatibility limitation of CEPE.
>
> In summary, **the differences in the encoder used and the number of trainable parameters in CEPE** arise from its inherent design. In fact, this **highlights the advantages of E2LLM in terms of parameter efficiency and compatibility**.
>
> We hope our clarifications address your concerns. Should you have any further questions, please do not hesitate to ask, and we would be happy to provide further explanations.

---

> > ### Author Response · Authors · 2024-12-03
> > **Additional Clarifications on CEPE and E2LLM for Reviewer JbEJ**
> >
> > We appreciate the time you have taken to review our paper and read our rebuttal. Since we have not received a response to our previous reply, we would like to provide further clarification regarding your concerns.
> >
> > When **CEPE** was proposed, **it did not anticipate the possibility of a better text encoder that could seamlessly plug into their framework** in the future. As a result, it specialized the encoder, aiming to ensure that its vocabulary matches the decoder’s, **and trained the encoder from scratch**.
> >
> > Additionally, we would like to emphasize that **CEPE not only ensures that the text encoder remains trainable, but also adds an extra randomly initialized cross-attention layer in each transformer block of the LLM decoder**. These two factors together contribute to a large number of trainable parameters. It is important to note that **the number of trainable parameters is fixed, and is not only unrelated to the LoRA rank but also cannot be adjusted.**
> >
> > In summary, **we are comparing CEPE as a whole method, rather than only comparing our adapter with CEPE’s adapter** (the one used to align the text encoder and the LLM decoder).
> >
> > We hope this explanation addresses your concerns. If you have any further questions, please feel free to raise, and we would be happy to provide additional clarifications.

---

> > > ### Comment · Reviewer_JbEJ · 2024-12-03
> > >
> > > Thank you for your continued engagement and detailed responses to my concerns. Your clarifications regarding CEPE and E2LLM have addressed several points, but I still believe that certain aspects could be better clarified and explicitly incorporated into the main text to strengthen the paper. Here are my final thoughts and suggestions:
> > >
> > > 1. Clarification of Encoder Choices: While your explanation about the differences between CEPE and E2LLM's encoder choices is helpful, I would encourage you to explicitly highlight this distinction in the main text of the paper to make the performance comparison more meaningful. Specifically, please provide a clear justification for why different encoders were used for these two models in your experiments and discuss how this choice impacts the results. This will ensure that readers can better understand and fairly evaluate the performance differences regarding different framework designs.
> > >
> > > 2. Citations of Related Work: Out of the three related works I previously pointed out, two are about positional embeddings. While you have cited the first paper (CLEX), I noticed that the second paper has not yet been cited. Including this citation, along with references to other relevant works in positional embeddings, would provide a more comprehensive context for readers. Adding such citations will also demonstrate the breadth of your engagement with the related literature.
> > >
> > > I appreciate the significant revisions and clarifications you have provided so far. If the points mentioned above, particularly the explicit inclusion of the encoder justification and additional citations, are reflected in the final version of the paper, I would be willing to raise my score.
> > >
> > > Thank you once again for your effort in addressing the feedback. I look forward to seeing the final version of your paper.

---

> > > > ### Author Response · Authors · 2024-12-03
> > > > **Acknowledgment of your feedback and contributions**
> > > >
> > > > Thank you for taking the time to carefully review our responses and for considering our work to be marginally above the acceptance threshold. We greatly appreciate your constructive feedback, which will undoubtedly enhance the quality of our work.
> > > >
> > > > We will follow your suggestions and incorporate the necessary revisions into the final version of the paper. Specifically:
> > > >
> > > > 1. **Clarification of Encoder Choices:** We will explicitly highlight the distinction between the encoder choices of CEPE and E2LLM in the main text. This will include a clear justification for why different encoders were used for these two models in our experiments, along with a discussion on how this choice impacts the results.
> > > > 2. **Citations of Related Work:** We will include the papers you mentioned, as well as other relevant works in this area. This will provide a more comprehensive context for the readers.
> > > >
> > > > Once again, we sincerely thank you for reviewing our paper and for your continued engagement in the rebuttal process. Your time and efforts have greatly contributed to the refinement and enhancement of our work. We look forward to submitting the final version of the paper, reflecting the improvements based on your insightful feedback.

---

### Official Review · Reviewer_t6Dh · 2024-11-01

**Soundness:** 3
**Presentation:** 4
**Contribution:** 2
**Rating:** 6
**Confidence:** 4

**Summary:**

In this paper, authors try to propose a framework to support LLM at long context side. They chunk the input text into different parts and try to compress then by an encoder and a LORA adaptor is used to improve the representation. They propose two training objectives to handle the understanding and reasoning at the same time. They conduct experiments on several benchmarks and show the effectiveness of their method.

**Strengths:**

1. The presentation is pretty clear to me and the T1, T2 and T3 categories make lot of sense and easy to follow. They summarize current methods in terms of them.
2. They do a comprehensive evaluation on their model to show the effectiveness.
3. The model aligns the encoder and decoder via a trainable adapter, minimizing the need for extensive retraining and making it easier to implement.

**Weaknesses:**

1. The encoder-decoder framework is reasonable but I am concerned that what if the LLM is a large scale one like llama 70B. will the train be feasible. Seems author just conductor experiments on small scales.
2. The chunk size is pretty heuristic and with different tasks, the performance varies with chunk size. Could we make it learnable as well?
3. Maybe another concern is the real time application, how do we plan for that given this framework.
4. With the encoder and adapter, how does the inference speed compared to the baseline?
5. Can we do the eval on RULER?

**Questions:**

Please refer to the above.

---

> ### Author Response · Authors · 2024-11-28
> **Response to Reviewer t6Dh (Part 1)**
>
> We sincerely appreciate the insightful feedback provided by the reviewer. Incorporating their comments has greatly enhanced the clarity and depth of our paper, thereby improving its overall quality. To address each point comprehensively, we have prepared a detailed response for every comment raised. In our response, comments from the reviewer are highlighted in italics, with our responses directly following. For clarity, quotations from the adjusted manuscript are presented in markdown quotation mode. The corresponding modifications in the paper are **highlighted in blue**. Unless otherwise stated in our response, all pages, equations, sections, and bibliographical references refer to those in the revised paper._
>
>
> _W1 - The encoder-decoder framework is reasonable but I am concerned that what if the LLM is a large scale one like llama 70B. will the train be feasible. Seems author just conductor experiments on small scales._
>
>
>
> Thank you for your valuable feedback. **E2LLM can indeed scale to larger language models, such as Llama2-70B.** We have attempted to use Llama2-70B as the decoder and evaluated its performance on QMSum. The results are presented in Appendix H.1 (Lines 1232-1259 on Page 23), as follows:
>
>
>
> > We adopt Llama2-70B as the decoder to further validate the feasibility of E2LLM on larger-scale language models (denoted as E2LLM-70B). During training, we apply 4-bit quantization using QLoRA's Parameter-Efficient-Finetuning (PEFT) method. We conduct training and evaluating on QMSum, assessing its performance using the R1, R2, RL, G-mean, and PPL metrics as well as their relative improvement over those corresponding to the 7B model. The results are shown in Table 10.
>
>
> Table 10: Performance on E2LLM with larger-scale model.
> |  | R1 | R2 | RL | G-mean | PPL |
> | --- | :---: | :---: | :---: | :---: | :---: |
> | E2LLM-7B | 0.2537 | 0.0655 | 0.1875 | 0.1461 | 13.68 |
> | E2LLM-70B | 0.2561 | 0.0652 | 0.2312 | 0.1569 | 11.98 |
> | Rel. Improv.  | +0.95% | -0.458% | +21.99% | +7.39% | +12.43% |
>
>
> > As shown in the table, the performance of E2LLM significantly improves when using Llama2-70B, particularly in terms of Rouge-L and PPL. It is important to note that Rouge-1 and Rouge-2 evaluate unigram and bigram overlaps, respectively, measuring the match between the generated text and reference text at the word and phrase levels. In contrast, **Rouge-L evaluates the similarity of the generated and reference texts based on the longest common subsequence (LCS), which measures structural similarity at the sentence level. This indicates that by leveraging a larger model, E2LLM is able to better capture the overall sentence structure and word order.** Additionally, the reduction in PPL further demonstrates the model’s ability to generate more coherent and reasonable content.

---

> ### Author Response · Authors · 2024-11-28
> **Response to Reviewer t6Dh (Part 2)**
>
> _W2 - The chunk size is pretty heuristic and with different tasks, the performance varies with chunk size. Could we make it learnable as well?_
>
>
>
> According to our ablation study on chunk size in Table 11 and Appendix H.2 (Page 24, Lines 1290-1297), we find that
>
> > **the differences in performance metrics across different chunk sizes are relatively small for all datasets used in this study, indicating that the alignment process in E2LLM can effectively mitigate the impact of chunk size on performance.** Nonetheless, selecting an optimal chunk size can still provide a slight performance boost.
>
>
>
> On the other hand, we acknowledge that making the chunk size learnable is interesting and is beneficial to the performance of E2LLM. To this end, we provided a discussion on learnable chunk size in Appendix I.2 (Page 25, Lines 1326-1343) as:
>
> > While the performance of E2LLM is not highly sensitive to chunk size, selecting the optimal size can enhance its effectiveness, as illustrated in Table 11. This raises an intriguing question: Can we determine a chunk size that further boosts the performance of E2LLM? We believe there are two primary approaches to achieving this goal.
>
>
>
>
> > First, **we can apply techniques commonly used to optimize hyperparameters in neural networks or during neural architecture search (NAS) to the chunk size learning process.** Approaches such as Bayesian optimization, reinforcement learning, and meta-learning can be adapted to optimize both the chunk size and E2LLM model parameters simultaneously.
>
>
> > Second, **chunk size can also be optimized independently of the E2LLM model parameters.** One promising strategy is to explore chunking based on the semantic relationships between tokens, such as through meta-chunking (Zhao et al., 2024). However, a significant challenge with existing chunking methods is that the tokens within a chunk only capture information from that specific chunk, resulting in a loss of contextual information from nearby chunks. To address this issue, we can consider the "late" chunking method (Gunther et al., 2024}. This approach first embeds all tokens of a long text using a long-context embedding model and then applies chunking by mean pooling the token embeddings within each chunk. While this method provides chunk embeddings that encapsulate full contextual information, it comes with the drawback of increased complexity, as the computational demands of text encoders scale quadratically with the length of the input text.
>
> _W3 - Maybe another concern is the real time application, how do we plan for that given this framework._
>
>
>
> Thanks for pointing this out! We notice that **CPT (continue-pretraining) followed by SFT (supervised fine-tuning)** i**s a prevalent practice to achieve practical long-context LLMs, as seen in the Llama and Qwen series**. As such, **we provide a discussion in Appendix I.1 (Lines 1303-1323 on Page 25) regarding how CPT can be conducted within the E2LLM framework** as follows:
>
> > While E2LLM demonstrates comparable or superior performance compared to various baseline methods when utilizing the same amount of fine-tuning data, it has become increasingly common to engage in continue-pretraining (CPT) alongside supervised fine-tuning (SFT). This approach typically involves leveraging substantial quantities of high-quality, long-context data. By doing so, systems can achieve a long-text language model (LLM) that exhibits versatility across diverse long-text tasks. This practice has been effectively illustrated by recent models in the Llama and Qwen series (Dubey et al., 2024; Yang et al., 2024; Hui et al., 2024).
>
>
>
> > In this context, we will explore the methodology for conducting CPT within the E2LLM framework. Given that E2LLM functions akin to an encoder-decoder architecture, it is logical to adopt pretraining tasks prevalent in other established encoder-decoder frameworks, such as T5 (Raffel et al., 2019), BART (Lewis et al., 2020), and GLM (Du et al., 2022). However, **considering the unique characteristics of the E2LLM model, we specifically recommend the use of prefix language modeling (PLM) (Wang et al., 2022) as the pretraining task.** This choice ensures that the CPT process aligns seamlessly with the subsequent SFT process.
>
>
>
> > Concretely, suppose that the chunk size of the text encoder is $C$ and the length of the decoder is $L$. In this setup, we can create random pretraining sequences with lengths of $\ell C + L - \ell$, where $1 \leq \ell \leq L-1$ represents the number of chunks. We then can partition the prefix segments of length $\ell C$ into $\ell$ individual chunks, which are then fed into the text encoder. The task for E2LLM during this CPT phase is to predict the remaining segments of length $L - \ell$. This structured approach not only enhances the model's ability to comprehend and generate lengthy texts but also sets a solid foundation for effective fine-tuning on targeted applications thereafter.

---

> ### Author Response · Authors · 2024-11-28
> **Response to Reviewer t6Dh (Part 3)**
>
> _W4 - With the encoder and adapter, how does the inference speed compared to the baseline?_
>
>
> Thank you for pointing that out. In fact, we conduct experiments related to inference efficiency, as discussed in Section 4.4 (Lines 507-511 on Page 10) and Appendix G on Page 21-22.
>
> > As displayed in Figures 4b and 4c, **E2LLM stands out with impressive results, demonstrating the lowest runtime and memory usage, especially for lengthy sequences at 73K.** This efficiency is primarily due to its relatively high compression ratio of approximately 100 times, which dramatically reduces the number of chunk tokens processed by the LLM decoder to a size much smaller than the original number of text tokens.

---

> ### Author Response · Authors · 2024-11-28
> **Response to Reviewer t6Dh (Part 4)**
>
> *W5 - Can we do the eval on RULER?*
>
>
>
> Thank you for your insightful suggestions. In response, we have incorporated experimental results on the RULER benchmark, as presented in Table 8. Furthermore, we have included a detailed discussion of the model's performance on the RULER benchmark to provide a more comprehensive analysis. As shown in Appendix F, Line 1076-1109 on Page 20-21.
>
>
>
> Table 8: Performance on RULER Benchmark. The best results are in bold, the second are underlined, and the third are wavy underlined.
>
> | Contex Length | 4K   ||| | 8K |||   | 16K |||  |
> | ------------- | ----- | ----- | ----- | ----- | ---- | ----- | ----- | ----- | ---- | ----- | ----- | ----- |
> | Task          | VT    | CWE   | FWE   | QA    | VT   | CWE   | FWE   | QA    | VT   | CWE   | FWE   | QA    |
> | LLama2-7B     | 27.00 | 85.60 | 74.33 | 63.00 | -    | -     | -     | -     | -    | -     | -     | -     |
> | LongLoRA      | 1.60  | 16.60 | 9.33  | 55.50 | 2.20 | 13.40 | 10.33 | 44.00 | 2.00 | 5.80  | 4.00  | 52.00 |
> | YaRN          | 19.80 | 15.20 | 20.33 | 57.00 | 1.80 | 10.30 | 11.67 | 34.50 | 1.40 | 3.90  | 5.33  | 29.00 |
> | LongLLMLingua | 5.20  | 7.60  | 44.67 | 14.50 | 4.20 | 5.70  | 24.33 | 16.0  | 7.00 | 2.00  | 27.33 | 15.50 |
> | LLoCO         | 0.00  | 27.70 | 24.67 | 32.50 | 0.00 | 24.10 | 17.00 | 28.50 | 0.00 | 20.90 | 22.67 | 20.00 |
> | E2LLM         | 0.00  | 15.60 | 21.33 | 40.50 | 0.00 | 14.30 | 18.67 | 37.00 | 0.00 | 16.30 | 19.33 | 37.50 |
> | **Contex Length** | **32K** |||  | **64K** |||  | **128K** ||| |
> | Task          | VT    | CWE   | FWE   | QA    | VT   | CWE   | FWE   | QA    | VT   | CWE   | FWE   | QA    |
> | LLama2-7B     | -     | -     | -     | -     | -    | -     | -     | -     | -    | -     | -     | -     |
> | LongLoRA      | 0.40  | 1.80  | 1.670 | 33.50 | OOM  | OOM   | OOM   | OOM   | OOM  | OOM   | OOM   | OOM   |
> | YaRN          | 1.20  | 2.80  | 2.00  | 28.50 | OOM  | OOM   | OOM   | OOM   | OOM  | OOM   | OOM   | OOM   |
> | LongLLMLingua | 6.20  | 0.30  | 11.33 | 18.50 | 5.20 | 0.30  | 13.33 | 15.0  | 5.20 | 0.40  | 21.67 | 4.50  |
> | LLoCO         | 0.00  | 0.10  | 24.00 | 4.50  | 0.00 | 2.40  | 15.67 | 9.00  | 0.00 | 3.30  | 4.33  | 2.00  |
> | E2LLM         | 0.00  | 3.50  | 16.67 | 28.00 | 0.00 | 4.90  | 13.33 | 16.50 | 0.00 | 2.50  | 8.67  | 7.50  |
>
> > We do not consider the retrieval tasks here, as they can be considered variants of the Needle-in-a-Haystack test. For the VT task, we set the number of variable name-binding chains and the number of times binding variable names in each chain to be 1 and 4, respectively. For the CWE and FWE tasks, we set the frequency of ten common words to be 30, uncommon words to be 3, and alpha as 2.0. Finally, for the QA task, we use two single-hop short-context QA datasets SQuAD and HotPotQA. For models that requires training, we reuse the checkpoints trained in Section 4.2.
>
>
>
> > The results are listed in Table 8. Given the diversity of tasks presented in RULER, we can clearly identify the strengths and weaknesses of each baseline method. Although Yarn and LongLoRA perform relatively well in the QA task, they struggle significantly with the CWE and FWE tasks. This is likely due to an attention distraction problem, which hampers their ability to focus on specific common or frequent words. Additionally, both methods encounter out-of-memory issues when the context length exceeds or equals 64K; for reference, we utilized an A100 GPU with 80GB of memory for inference. This suggests that the space complexity of YaRN and LongLoRA is too high for scenarios with limited resources. On the other hand, LongLLMLingua excels in the FWE task but underperforms in the others. The soft compression methods, E2LLM and LLoCO, manage to strike a balance between performance on the aggregation (CWE and FWE) and QA tasks, yielding comparable results. E2LLM tends to favor QA tasks, while LLoCO is better suited for aggregation tasks. It is worth noting that E2LLM can take advantage of increasingly sophisticated text encoders that are continuously being open-sourced, as demonstrated in our ablation studies; meanwhile, the encoder used by LLoCO is fixed to AutoCompressor. Lastly, we observe that all methods perform poorly on the VT task, which demands a nuanced comprehension of the long context, presenting a challenge that may be too great for the current models.

---

> ### Author Response · Authors · 2024-12-02
> **Seeking for your further feedback**
>
> Dear Reviewer t6Dh,
>
>
> We would like to thank you once again for your constructive feedback on our paper. We have made significant efforts to address the concerns you raised and have revised the manuscript accordingly. Given that the discussion period is nearing its end, we kindly ask if you could review the updated version and let us know if there are any remaining issues or points that require further clarification.
>
> We are more than happy to address any additional questions or concerns you might have.
>
> Thank you for your time and consideration.

---

> > ### Comment · Reviewer_t6Dh · 2024-12-03
> >
> > Thanks for your response and it resolved my concerns. It would be great to include them in the next version of your paper. I will raise my score accordingly. thanks!

---

> > > ### Author Response · Authors · 2024-12-03
> > > **Acknowledgment of your feedback and contributions**
> > >
> > > We sincerely appreciate your acknowledgment that our additional experiments and clarifications have successfully addressed your concerns. Once again, we are truly grateful for your thoughtful feedback and invaluable insights, which have played a crucial role in refining and improving our work.

---

### Official Review · Reviewer_9zKM · 2024-11-03

**Soundness:** 2
**Presentation:** 2
**Contribution:** 3
**Rating:** 5
**Confidence:** 4

**Summary:**

The paper introduces E2LLM, a method for long-context language modeling. The main components consist of an encoder, an adapter, and a decoder-only LLM. It uses Llama 2 7B chat as the backbone LLM and GTE as the encoder. It’s evaluated on several long-context and QA tasks like QMSum, GovReport, Quality, NarrativeQA, and TriviaQA, and achieves competitive results. It’s also evaluated on the needle-in-a-haystack and LongBench.

**Strengths:**

- E2LLM is a novel method with significant inference benefits compared to previous works. It shows that encoder and decoder can be connected with an adapter in the middle.
- The method shows strong empirical results on downstream tasks in the supervised training settings., and includes a number of tasks in the evaluation setting.
- The paper also includes a number of ablation studies that reveal more insights regarding the method.

**Weaknesses:**

- About the baseline methods, Appendix B states that Llama 2 7B is “the backbone for the other methods”, but E2LLM uses Llama 2 7B Chat as the decoder. It doesn’t seem fair to compare methods that use different backbone models. What would the performance of E2LLM be if Llama 2 7B were used as the backbone model instead? Or if the other methods use the chat model as the backbone?
- A key difference between previous methods and E2LLM is that they are typically studied in the pre-training/continual pre-training settings where they are first trained on a corpus like RedPajama (such as in the case of YaRN, LongLoRA, CEPE, and others), and then evaluated either zero-shot or with ICL on downstream tasks. It would be important to see how E2LLM would perform in such settings in addition to the supervised fine-tuning setting.

On the writing, there are some citation errors, for example:
Line 033: Giorgi et al. → Giorgi et al., 2023.
Line 053: Ding et al. → Ding et al., 2024.
Line 090: Kenton & Toutanova, 2019 → Devlin et al., 2019.
There are others throughout the paper.

**Questions:**

In addition to the questions listed in the Weakness section:

- In Figure 4, why do the inference time and memory usage increase for StreamingLLM? It keeps a constant size of starting and recent KV caches, so it should stay constant past a certain length.
- Why is E2LLM better than YaRN and LLoCO on NarrativeQA in Table 1 but not in Table 2?
- In Sec 4.1, the datasets listed are QMSum, GovReport, HotpotQA, NarrativeQA, and TriviaQA, but the corresponding Table 4 lists Quality instead of HotpotQA for the statistics, and Table 1 shows Quality instead of HotpotQA.
- How important are the two training tasks in Sec 3.2? It's not clear how they are used during training and how they affect the performance and the training efficiency, as reconstructing the inputs from the encoder introduces additional costs.

---

> ### Author Response · Authors · 2024-11-28
> **Response to Reviewer 9zKM (Part 1)**
>
> We deeply value the thoughtful feedback received. Addressing the comments from the reviewer has significantly improved both the clarity and the depth of our paper, elevating its overall quality. To ensure a thorough response, we have prepared a detailed reply for each point noted by the reviewer. In our response, comments from the reviewer are highlighted in italics, with our responses directly following. For clarity, quotations from the adjusted manuscript are presented in markdown quotation mode. The corresponding modifications in the paper are **highlighted in blue**. Unless otherwise stated in our response, all pages, equations, sections, and bibliographical references refer to those in the revised paper.
>
>
>
> *W1 - About the baseline methods, Appendix B states that Llama 2 7B is “the backbone for the other methods”, but E2LLM uses Llama 2 7B Chat as the decoder. It doesn’t seem fair to compare methods that use different backbone models. What would the performance of E2LLM be if Llama 2 7B were used as the backbone model instead? Or if the other methods use the chat model as the backbone?*
>
>
>
> Thank you for your feedback. **For the other baselines, we use the same backbone as E2LLM, namely LLaMA2-7B-Chat.** As described in Appendix B (Page 15, Lines 796-797)，
>
> >  **LLaMA2-7B: This refers to LLaMA2-7B-Chat** without additional training or fine-tuning, serving as the backbone for the other methods.
>
> We realize that inconsistent naming may have caused some confusion. To enhance the readability of the paper, **we have standardized the naming throughout the manuscript, replacing all occurrences of "LLaMA2-7B-Chat" with "LLaMA2-7B" in relevant sections.**

---

> ### Author Response · Authors · 2024-11-28
> **Response to Reviewer 9zKM (Part 2)**
>
> *W2 - A key difference between previous methods and E2LLM is that they are typically studied in the pre-training/continual pre-training settings where they are first trained on a corpus like RedPajama (such as in the case of YaRN, LongLoRA, CEPE, and others), and then evaluated either zero-shot or with ICL on downstream tasks. It would be important to see how E2LLM would perform in such settings in addition to the supervised fine-tuning setting.*
>
>
>
> We would like to argue that **most long-context extension methods aim to enhance the context window with minimal to no fine-tuning**, as explicitly stated in studies like Position Interpolation (Chen et al. (2023a)) and YaRN (Peng et al., 2023). This is consistent with E2LLM’s goal of compatibility with pretrained models. As discussed in Section 1 (Lines 041-044),
>
> > Achieving this compatibility is crucial for effectively leveraging the pretrained knowledge contained in these models, allowing for **parameter and sample efficiency** without necessitating extensive additional training with large datasets.
>
> **This rationale underpins our decision not to focus on continue-pretraining (CPT) plus supervivsed fine tuning (SFT) in the first place, as it deviates from the original motivation behind long-context extension methods**. Our core aim is to demonstrate that, given an equivalent training dataset, E2LLM—being tuned with fewer parameters—can outperform baseline methods that necessitate more extensive tuning. This assertion is validated in Section 4 (see Tables 1-2 and Figure 3), where E2LLM effectively harnesses the knowledge embedded in the pretrained text encoder and LLM decoder.
>
>
>
> On the other hand, **we acknowledge that CPT plus SFT is a prevalent practice to achieve practical long-context LLMs, as seen in the Llama and Qwen series**. However, due to resource constraints and time limitations, we could not conduct CPT plus SFT using the E2LLM framework on a substantial amount of high-quality long-context data. Instead, **we provide a discussion in Appendix I (Lines 1303-1323 on Page 25) regarding how CPT can be conducted within the E2LLM framework** as follows:
>
> > While E2LLM demonstrates comparable or superior performance compared to various baseline methods when utilizing the same amount of fine-tuning data, it has become increasingly common to engage in continue-pretraining (CPT) alongside supervised fine-tuning (SFT). This approach typically involves leveraging substantial quantities of high-quality, long-context data. By doing so, systems can achieve a long-text language model (LLM) that exhibits versatility across diverse long-text tasks. This practice has been effectively illustrated by recent models in the Llama and Qwen series (Dubey et al., 2024; Yang et al., 2024; Hui et al., 2024).
>
>
>
> > In this context, we will explore the methodology for conducting CPT within the E2LLM framework. Given that E2LLM functions akin to an encoder-decoder architecture, it is logical to adopt pretraining tasks prevalent in other established encoder-decoder frameworks, such as T5 (Raffel et al., 2019), BART (Lewis et al., 2020), and GLM (Du et al., 2022). However, **considering the unique characteristics of the E2LLM model, we specifically recommend the use of prefix language modeling (PLM) (Wang et al., 2022) as the pretraining task.** This choice ensures that the CPT process aligns seamlessly with the subsequent SFT process.
>
>
>
> > Concretely, suppose that the chunk size of the text encoder is $C$ and the length of the decoder is $L$. In this setup, we can create random pretraining sequences with lengths of $\ell C + L - \ell$, where $1 \leq \ell \leq L-1$ represents the number of chunks. We then can partition the prefix segments of length $\ell C$ into $\ell$ individual chunks, which are then fed into the text encoder. The task for E2LLM during this CPT phase is to predict the remaining segments of length $L - \ell$. This structured approach not only enhances the model's ability to comprehend and generate lengthy texts but also sets a solid foundation for effective fine-tuning on targeted applications thereafter.

---

> ### Author Response · Authors · 2024-11-28
> **Response to Reviewer 9zKM (Part 3)**
>
> *Q1 - In Figure 4, why do the inference time and memory usage increase for StreamingLLM? It keeps a constant size of starting and recent KV caches, so it should stay constant past a certain length.*
>
>
>
> Indeed, StreamingLLM does maintain a fixed-size KV cache for both starting and recent tokens, which allows it to handle streaming inputs and outputs effectively. However, **as indicated in the official implementation, for any given context, StreamingLLM must initially load the entire KV cache associated with that context.** During the subsequent generation process, it utilizes Sink Attention to preserve the KV caches for both the starting and recent tokens. Consequently, in long-context scenarios, the memory usage and inference time for StreamingLLM still exhibit quadratic growth.
>
>
>
> We have addressed this point in Appendix G (Lines 1142-1147, Page 22) when discussing the results in Figure 4.
>
>
>
> *Q2 - Why is E2LLM better than YaRN and LLoCO on NarrativeQA in Table 1 but not in Table 2?*
>
>
>
> Thanks for pointing this out! The difference primarily arises from the experimental settings of Table 1 and Table 2. Specifically, in Table 1, each dataset is trained and tested independently. As mentioned in Section 4.1 (Page 7, Lines 341-342),
>
> > We employ the validation sets of each dataset for testing and split the training sets into training and validation subsets using a 95:5 ratio.
>
> In contrast, Table 2 involves training on a mixture of all five datasets. As stated in Section 4.2 (Page 8, Lines 421-422),
>
> > The methods that require training utilize data collected from all five tasks discussed in the previous subsection.
>
> Furthermore, Appendix E (Page 20, Lines 1071-1072) clarifies:
>
> > For methods that require training, the training data utilized are identical to those employed during the Needle in a Haystack experiment.
>
> We hypothesize that interactions between datasets can have varying effects, maybe some positive and others negative. This explains why E2LLM demonstrates better performance in the single-task experiments of Table 1 but falls behind some baselines in the multi-task setting of Table 2.
>
>
>
> *Q3 - In Sec 4.1, the datasets listed are QMSum, GovReport, HotpotQA, NarrativeQA, and TriviaQA, but the corresponding Table 4 lists Quality instead of HotpotQA for the statistics, and Table 1 shows Quality instead of HotpotQA.*
>
>
>
> Thank you for bringing this to our attention. We acknowledge that there was an error in Section 4.1 and Table 1, where **It should indeed be "Quality" rather than "HotpotQA"**. The introductions of the datasets in Appendix C, as well as the notations in Tables 4 and 5, are accurate. We have made the necessary corrections in the revised manuscript.
>
>
>
> *Q4 - How important are the two training tasks in Sec 3.2? It's not clear how they are used during training and how they affect the performance and the training efficiency, as reconstructing the inputs from the encoder introduces additional costs.*
>
>
>
> We have conducted an ablation study regarding the weight of the "understanding" task in Appendix H.2 (Page 24, Lines 1267-1275). Moreover, we have made it clear in Section 3.2 (Page 5, Lines 264-266) that
>
> > It is important to note that **the “understanding” task serves as an auxiliary task**, while our primary focus remains on the “reasoning” task. We determine the final checkpoints exclusively based on the validation loss associated with the “reasoning” task.
>
>
>
> Regarding the impact of the "understanding" task on training efficiency, we acknowledge that it does introduce some additional costs, as the training speed is influenced by the length of the context that must be reconstructed. Nevertheless, we contend that this task is essential. In the absence of continual pretraining, a self-supervised task like reconstruction or "understanding" is vital for maintaining the model’s ability to retain context. Without it, we observe a substantial degradation in performance. This phenomenon is illustrated in Figure 6 on Page 23, where we see a marked decline in performance as the weight of the "understanding" task approaches zero (equivalent to the logarithm of its weight approaching negative infinity).

---

> ### Author Response · Authors · 2024-12-02
> **Seeking for your further feedback**
>
> Dear Reviewer 9zKM,
>
> We would like to thank you once again for your constructive feedback on our paper. We have made significant efforts to address the concerns you raised and have revised the manuscript accordingly. Given that the discussion period is nearing its end, we kindly ask if you could review the updated version and let us know if there are any remaining issues or points that require further clarification.
>
> We are more than happy to address any additional questions or concerns you might have.
>
> Thank you for your time and consideration.

---

### Official Review · Reviewer_6Bzq · 2024-11-04

**Soundness:** 3
**Presentation:** 1
**Contribution:** 3
**Rating:** 6
**Confidence:** 3

**Summary:**

The paper presents E2LLM, an architecture that addresses long-context processing in LLMs through a chunk-based approach.

At its core, E2LLM employs a pretrained text encoder to compress text chunks into embedding vectors, which are then aligned with a decoder-only LLM (LLaMA-2) through a two-layer MLP adapter.

E2LLM also propose a two-stage training process: 1. an "understanding" task where the model reconstructs the original text from compressed representations, and 2. a "reasoning" task focused on answering queries using these compressed contexts.

The authors position their work as solving the "impossible triangle"(I am not so sure about the reference here) of maintaining strong performance, computational efficiency, and compatibility with pretrained models.

I am concerned with unclear explanations of the core model architectures, not well collected benchmarks, and lack of in-depth analysis

**Strengths:**

1. Solving long-context language modeling is an important problem

2. Compression yields benefits in computation efficiency

3. In their selected benchmarks, their model yield good performance.

**Weaknesses:**

## Benchmark results are not tailored for the long context length argued in the paper.

As far as I understand, most of benchmarks stated in this paper, their AVG length is not up-to-128k. And benchmarks are not on par with the recent advances, for example, RULER where there are way more challenging tasks mentioned in recent papers for example HELMET[1] and long context controlled study[3]

-  I would encourage authors to break down the task accuracy by length to show the real benefit of their method.

- Also adding some recent benchmarks, for example Ruler[1], and HELMET[2] to have real long context evaluation thoroughly.

- I think the Needle task is not well presented. I understand that authors wanna focus on the accuracy, but the heat map might give us a clear image on where the model fails, and the trend per position more clearly w.r.t needle depth. Is it possible to have a similar analysis?


## Presentation can be improved.

- I am not so sure that I understand the compression process. The major compression is depending on the chunk size in this paper, but it is not well studies how this chunk size will influence the results, or how will influence the results, semantically for example. I would encourage authors break down the Negative LL per token position to demonstrate the effect of chunk size, particularly at boundaries.

- Model architecture details are missing. Maybe this can be benefit from adding a clear model architecture figure and justifying the design choice. Figure 1 demonstrates the idea at very high level and paragraph descriptions are not very clear in 3.1.

References

[1] RULER: What's the Real Context Size of Your Long-Context Language Models?

[2] HELMET: How to Evaluate Long-Context Language Models Effectively and Thoroughly

[3] A Controlled Study on Long Context Extension and Generalization in LLMs

**Questions:**

Q1. What's the adapter is chosen and why?

Q2. How the chunk size will affect results


See weakness for most of the comments and questions.

---

> ### Author Response · Authors · 2024-11-28
> **Response to Reviewer 6Bzq (Part 1)**
>
> We're grateful for your insightful comments! We believe that addressing the reviewer’s comments has resulted in improving the clarity and presentation of the paper’s contributions and has brought the paper to a higher standard. A comprehensive reply has been prepared to address every point raised by the reviewer. The reviewer's comments are presented in italics, followed by our response. Quotations from the revised paper are included in markdown quotation mode. The corresponding modifications in the paper are **highlighted in blue**. Unless otherwise stated in our response, all pages, equations, sections, and bibliographical references refer to those in the revised paper.
>
>
>
> *W1 - I would encourage authors to break down the task accuracy by length to show the real benefit of their method.*
>
>
>
> Thank you for your valuable suggestions. Following your advice, we conduct an evaluation on the QMSum and NarrativeQA datasets based on context length to gain deeper insights. **The results are presented in Appendix C on Page 18** as follows:
>
> > Next, we investigate the sensitivity of the models' performance to variations in context length. To do this, we categorize samples from the QMsum and NarrativeQA datasets into five groups based on their context lengths and then evaluate the perplexity (PPL) of the answers within each group. Our findings are summarized in Table 6.
>
> Table 6-1: Performance as a function of context length. The best results are in bold, the second are underlined, and the third are wavy underlined.
>
> | Methods       | 0k-6k   |  | 6k-12k  |  | 12k-18k |  | 18k-24k|   | 24k+    |  |
> | ------------- | --------- | --------- | --------- | --------- | --------- | --------- | --------- | --------- | --------- | --------- |
> || G-mean        | PPL       | G-mean    | PPL       | G-mean    | PPL       | G-mean    | PPL       | G-mean    | PPL       |
> | Llama2-7B     | 13.05     | 28.57     | 11.99     | 85.35     | 11.54     | 84.31     | 12.56     | 81.74     | 10.32     | 85.60     |
> | StreamingLLM  | 3.27      | 36.35     | 4.21      | 168.63    | 3.32      | 224.24    | 3.26      | 356.17    | 2.45      | 362.41    |
> | LongLoRA      | 5.91      | 12.92     | 8.13      | 13.17 | 8.3       | 14.65     | 9.66      | 15.97     | 7.44      | 17.31     |
> | CEPE          | 11.66     | 128.01    | 10.42     | 144.34    | 9.29      | 161.28    | 8.21      | 145.54    | 6.56      | 234.24    |
> | YaRN          | 13.57     | 14.52     | 12.10     | 14.02     | 12.88     | 17.06     | 11.49     | 17.75     | 6.33      | 18.90     |
> | RAG           | 6.12      | 17.94     | 8.72      | 17.58     | 9.65      | 20.95     | 9.03      | 19.59     | 6.24      | 19.39     |
> | LongLLMLingua | 7.73      | 11.25 | 9.83      | 15.12     | 8.72      | 16.25     | 9.08      | 19.66     | 8.87      | 21.55     |
> | LLoCO         | 13.63     | 34.56     | 12.78     | 41.27     | 13.15     | 47.45     | 12.13     | 47.87     | 10.03     | 56.30     |
> | E2LLM         | 15.04 | 12.69     | 15.27 | 13.47     | 14.14 | 13.95 | 14.26 | 13.33 | 15.31 | 13.92 |

---

> ### Author Response · Authors · 2024-11-28
> **Response to Reviewer 6Bzq (Part 2)**
>
> Table 6-2: Performance as a function of context length. The best results are in bold, the second are underlined, and the third are wavy underlined.
>
> | Methods       | 0k-24k |   | 24k-48k |  | 48k-72k |  | 72k-96k |  | 96k+  |  |
> | ------------- | --------- | --------- | --------- | --------- | --------- | --------- | --------- | --------- | --------- | --------- |
> || F1            | PPL       | F1        | PPL       | F1        | PPL       | F1        | PPL       | F1        | PPL       |           |
> | Llama2-7B     | 3.10      | 75.81     | 10.71     | 178.28    | 7.51      | 250.81    | 0.61      | 2303.08   | 2.48      | 2215.08   |
> | StreamingLLM  | 4.36      | 79.34     | 2.53      | 135.71    | OOM       | OOM       | OOM       | OOM       | OOM       | OOM       |
> | LongLoRA      | 3.23      | 11.93 | 9.47      | 12.17 | OOM       | OOM       | OOM       | OOM       | OOM       | OOM       |
> | CEPE          | 3.37      | 3568.12   | 2.65      | 2272.04   | OOM       | OOM       | OOM       | OOM       | OOM       | OOM       |
> | YaRN          | 7.19      | 13.94     | 6.59      | 17.16     | OOM       | OOM       | OOM       | OOM       | OOM       | OOM       |
> | RAG           | 2.40      | 12.98     | 2.14      | 41.35     | 2.55      | 60.28     | 2.14      | 58.32     | 1.43      | 57.20     |
> | LongLLMLingua | 7.84      | 26.52     | 6.23      | 29.45     | 3.16      | 29.96     | 1.72      | 38.53     | 1.03      | 48.53     |
> | LLoCO         | 10.89     | 13.32     | 10.67     | 15.67     | 10.88     | 17.31     | 11.42     | 16.19     | 9.43      | 18.54     |
> | E2LLM         | 12.12 | 13.45     | 12.41 | 12.87     | 12.76 | 12.96 | 12.23 | 13.65 | 11.97 | 13.71 |
>
> > The results presented in the table indicate that **E2LLM demonstrates a strong resilience to variations in context length for both summarization (QMSum) and question-answering (NarrativeQA) tasks, typically achieving the best results among all models.** This robustness can be attributed to the "understanding" task incorporated during the training of E2LLM (see Section 3.2). By reconstructing different parts of the context, E2LLM effectively comprehends the information, regardless of its length.
>
> > Notably, the performances of YaRN, LongLoRA, CEPE, RAG, and LongLLMLingua also exhibit insensitivity to context length. On the other hand, LLoCO's performance declines slowly with increasing context length. Finally, streamingLLM and the original Llama2-7B demonstrate sensitivity to context length; streamingLLM loses more information in the middle of the context as length increases due to its specific $\Lambda$-shaped attention mask, while Llama2-7B struggles to handle long contexts altogether, as its maximum length has not been extended.

---

> ### Author Response · Authors · 2024-11-28
> **Response to Reviewer 6Bzq (Part 3)**
>
> *W2 - Also adding some recent benchmarks, for example Ruler[1], and HELMET[2] to have real long context evaluation thoroughly.*
>
>
> Thank you for your insightful suggestions. In response, we have incorporated experimental results on the RULER benchmark, as presented in Table 8. Furthermore, we have included a detailed discussion of the model's performance on the RULER benchmark to provide a more comprehensive analysis. As shown in Appendix F, Line 1076-1109 on Page 20-21.
>
>
>
> Table 8: Performance on RULER Benchmark. The best results are in bold, the second are underlined, and the third are wavy underlined.
>
> | Contex Length | 4K   ||| | 8K |||   | 16K |||  |
> | ------------- | ----- | ----- | ----- | ----- | ---- | ----- | ----- | ----- | ---- | ----- | ----- | ----- |
> | Task          | VT    | CWE   | FWE   | QA    | VT   | CWE   | FWE   | QA    | VT   | CWE   | FWE   | QA    |
> | LLama2-7B     | 27.00 | 85.60 | 74.33 | 63.00 | -    | -     | -     | -     | -    | -     | -     | -     |
> | LongLoRA      | 1.60  | 16.60 | 9.33  | 55.50 | 2.20 | 13.40 | 10.33 | 44.00 | 2.00 | 5.80  | 4.00  | 52.00 |
> | YaRN          | 19.80 | 15.20 | 20.33 | 57.00 | 1.80 | 10.30 | 11.67 | 34.50 | 1.40 | 3.90  | 5.33  | 29.00 |
> | LongLLMLingua | 5.20  | 7.60  | 44.67 | 14.50 | 4.20 | 5.70  | 24.33 | 16.0  | 7.00 | 2.00  | 27.33 | 15.50 |
> | LLoCO         | 0.00  | 27.70 | 24.67 | 32.50 | 0.00 | 24.10 | 17.00 | 28.50 | 0.00 | 20.90 | 22.67 | 20.00 |
> | E2LLM         | 0.00  | 15.60 | 21.33 | 40.50 | 0.00 | 14.30 | 18.67 | 37.00 | 0.00 | 16.30 | 19.33 | 37.50 |
> | **Contex Length** | **32K** |||  | **64K** |||  | **128K** ||| |
> | Task          | VT    | CWE   | FWE   | QA    | VT   | CWE   | FWE   | QA    | VT   | CWE   | FWE   | QA    |
> | LLama2-7B     | -     | -     | -     | -     | -    | -     | -     | -     | -    | -     | -     | -     |
> | LongLoRA      | 0.40  | 1.80  | 1.670 | 33.50 | OOM  | OOM   | OOM   | OOM   | OOM  | OOM   | OOM   | OOM   |
> | YaRN          | 1.20  | 2.80  | 2.00  | 28.50 | OOM  | OOM   | OOM   | OOM   | OOM  | OOM   | OOM   | OOM   |
> | LongLLMLingua | 6.20  | 0.30  | 11.33 | 18.50 | 5.20 | 0.30  | 13.33 | 15.0  | 5.20 | 0.40  | 21.67 | 4.50  |
> | LLoCO         | 0.00  | 0.10  | 24.00 | 4.50  | 0.00 | 2.40  | 15.67 | 9.00  | 0.00 | 3.30  | 4.33  | 2.00  |
> | E2LLM         | 0.00  | 3.50  | 16.67 | 28.00 | 0.00 | 4.90  | 13.33 | 16.50 | 0.00 | 2.50  | 8.67  | 7.50  |
>
> > We do not consider the retrieval tasks here, as they can be considered variants of the Needle-in-a-Haystack test. For the VT task, we set the number of variable name-binding chains and the number of times binding variable names in each chain to be 1 and 4, respectively. For the CWE and FWE tasks, we set the frequency of ten common words to be 30, uncommon words to be 3, and alpha as 2.0. Finally, for the QA task, we use two single-hop short-context QA datasets SQuAD and HotPotQA. For models that requires training, we reuse the checkpoints trained in Section 4.2.
>
> > The results are listed in Table 8. Given the diversity of tasks presented in RULER, we can clearly identify the strengths and weaknesses of each baseline method. Although Yarn and LongLoRA perform relatively well in the QA task, they struggle significantly with the CWE and FWE tasks. This is likely due to an attention distraction problem, which hampers their ability to focus on specific common or frequent words. Additionally, both methods encounter out-of-memory issues when the context length exceeds or equals 64K; for reference, we utilized an A100 GPU with 80GB of memory for inference. This suggests that the space complexity of YaRN and LongLoRA is too high for scenarios with limited resources. On the other hand, LongLLMLingua excels in the FWE task but underperforms in the others. The soft compression methods, E2LLM and LLoCO, manage to strike a balance between performance on the aggregation (CWE and FWE) and QA tasks, yielding comparable results. E2LLM tends to favor QA tasks, while LLoCO is better suited for aggregation tasks. It is worth noting that E2LLM can take advantage of increasingly sophisticated text encoders that are continuously being open-sourced, as demonstrated in our ablation studies; meanwhile, the encoder used by LLoCO is fixed to AutoCompressor. Lastly, we observe that all methods perform poorly on the VT task, which demands a nuanced comprehension of the long context, presenting a challenge that may be too great for the current models.

---

> ### Author Response · Authors · 2024-11-28
> **Response to Reviewer 6Bzq (Part 4)**
>
> *W3- I think the Needle task is not well presented. I understand that authors wanna focus on the accuracy, but the heat map might give us a clear image on where the model fails, and the trend per position more clearly w.r.t needle depth. Is it possible to have a similar analysis?*
>
>
>
> We would like to clarify that we have included a heat map in Figure 3 on Page 9 and have discussed its implications in Section 4.2.
>
>
>
> Following your suggestion, **we have further plotted the average score as a function of the depth percent in Figure 5 and discussed the results in Appendix D (Lines 1014-1019)** as:
>
> > In addition to the results presented in Section 4.2, we further illustrated the average score over the context length as a function of depth percentage in Figure 5. It is evident that the performances of Llama2-7B, YaRN, LongLoRA, LongLLMLingua, and E2LLM are largely insensitive to the depth at which the needle is inserted. In contrast, LLoCO achieved the best results when the needle was positioned close to the answer, as discussed at the end of Section 4.2. Furthermore, E2LLM typically delivers the best performance across all depths.
>
>
>
> *W4 & Q2 - I am not so sure that I understand the compression process. The major compression is depending on the chunk size in this paper, but it is not well studies how this chunk size will influence the results, or how will influence the results, semantically for example. I would encourage authors break down the Negative LL per token position to demonstrate the effect of chunk size, particularly at boundaries.*
>
>
>
> **We have indeed examined the impact of chunk size on the performance of E2LLM, as detailed in Table 10 on Page 24 and further discussed in Appendix H.2** (Lines 1290-1298) as:
>
> > Results in Table 11 show that **the differences in performance metrics across different chunk sizes are relatively small for all datasets used in this study, indicating that the alignment process in E2LLM can effectively mitigate the impact of chunk size on performance.** Nonetheless, **selecting an optimal chunk size can still provide a slight performance boost.** While smaller chunks might reduce compression and better preserve inputs, they may hinder context capture in longer sentences or paragraphs, making it difficult for the encoder to grasp semantics, which affects downstream tasks. Conversely, larger chunk sizes increase diversity and noise, complicating semantic capture and leading to decreased performance, especially in tasks like DocumentQA where relevant sentences may be overlooked.
>
>
>
> Additionally, we would like to clarify that **we are unable to compute the negative log-likelihood per token position for the input context in E2LLM. This is because the loss in E2LLM is calculated solely based on the answers generated by the LLM decoder, rather than on the context provided to the text encoder.** The context serves merely as a prefix in the E2LLM framework.
>
>
>
> We appreciate your feedback and hope that our explanations address your concerns regarding the chunk size and its effects on model performance.

---

> ### Author Response · Authors · 2024-11-28
> **Response to Reviewer 6Bzq (Part 5)**
>
> *W5 - Model architecture details are missing. Maybe this can be benefit from adding a clear model architecture figure and justifying the design choice. Figure 1 demonstrates the idea at very high level and paragraph descriptions are not very clear in 3.1.*
>
>
>
> Thanks for pointing this out! We would like to clarify that **we provided the model architecture in Figure 2 on Page 2.**
>
>
>
> We also identified an error in the previous version of Section 3.1; it should reference "Figure 2" as the illustration of the E2LLM framework architecture instead of "Figure 1." This mistake has been rectified in the revised manuscript.
>
>
>
> Furthermore, **we have enhanced Section 3.1 with a more comprehensive description of the model architecture as:**
>
>
>
> > For long input contexts, E2LLM first performs chunking. Each resulting chunk is then processed by the encoder, which captures its semantics. The adapter facilitates the mapping of the encoder's outputs into the LLM decoder’s embedding space, allowing the decoder to interpret these representations effectively. Ultimately, the decoder utilizes these embeddings as substitutes for the original context and executes two fine-tuning tasks—“understanding” and “reasoning”—to train the entire framework.
>
>
>
> *Q1 - What's the adapter is chosen and why?*
>
>
>
> As mentioned in **Section 3.1 (Page 5, Lines 236-237) and Appendix B (Page 16-17, Lines 862-863, 874-876)**,
>
> > Regarding the Adapter, its structure is designed as a two-layer MLP. The first layer's input and output neuron numbers correspond to the embedding dimensions of the encoder and decoder, respectively, with GELU used as the activation function. The second layer maintains equal input and output dimensions, aligning with the decoder's embedding size.
>
>
>
> Our decision is informed by inspiration drawn from vision-language models (VLM), as discussed in Section 3.3 (Page 6). In the literature, we identified three primary adapter designs: the linear layer employed in MiniGPT-4 (Zhu et al., 2024), the MLP used in LLaVA (Liu et al., 2024), and the cross-attention layer with learnable queries found in Qwen-VL (Bai et al., 2023), which was later replaced by an MLP in Qwen2-VL ([R1]).
>
>
>
> **We opted for a simple yet effective adapter design**. Our ablation study, outlined in Appendix H.2 (Lines 1284-1287), reveals that a single linear layer falls short when compared to the two-layer MLP; this is largely because **a linear transformation may inadequately map the output of the text encoder to the embedding space of the LLM decoder**. Additionally, we have identified significant limitations with **cross-attention layers**. Specifically, these adapters tend to convert an arbitrary number of input chunks into a fixed number of learnable queries, regardless of the context length. This approach **seems impractical in our scenario as it fails to account for the variability in context length.**
>
>
>
> [R1]  Peng Wang et al., Qwen2-VL: Enhancing Vision-Language Model's Perception of the World at Any Resolution. arXiv 2024.

---

> > ### Comment · Reviewer_6Bzq · 2024-11-30
> > **response**
> >
> > thank the authors for the clarification. I don't have further questions. In light of those changes, I have changed my score to 6.

---

> > > ### Author Response · Authors · 2024-11-30
> > > **Acknowledgment of your feedback and contributions**
> > >
> > > We are truly grateful for your recognition of that our additional experiments and clarifications have effectively addressed your concerns. Once more, we deeply appreciate your thoughtful feedback and invaluable insights, which have greatly contributed to the refinement and enhancement of our work.

---

### Author Response · Authors · 2024-11-28
**Response to All Reviewers**

We would like to express our sincere gratitude to all the reviewers for their insightful feedback and constructive suggestions, which have greatly enhanced the quality of our work. Below, we provide a summary of the revisions and additional experiments carried out to address the reviewers' concerns:

In response to the reviewers’ comments, we have conducted **6 new experiments and included 5 additional tables and 1 new figure** to further support and validate our claims. These updates include:

1. **Table 6**: We bucket the evaluation datasets based on context length and assess the metrics and PPL of E2LLM and the baselines, to validate the performance variations across different context lengths, gaining deeper insights.
2. **Table 8**: We add a new benchmark RULER to validate the performance of E2LLM, providing a more comprehensive analysis.
3. **Figure 5**: We include the model's accuracy as a function of depth percent in the Needle In A Haystack task, further verifying the impact of depth on needle retrieval.
4. **Table 10**: We perform experiments with Llama2-70B as the decoder model to validate the feasibility of training E2LLM on large-scale language models.
5. **Table R1**: We add the CLEX baseline and evaluate its metrics and PPL to further assess the performance of E2LLM in long-context scenarios.
6. **In response to Reviewer uKEN**, we conduct experiments under different chunk sizes, where E2LLM is tasked with reconstructing the original context. We report Precision, Recall, and F1 scores to investigate the impact of compression ratio on reconstruction ability.

---

### Meta-Review · Area_Chair_Csq4 · 2024-12-18

**Metareview:**

This paper proposes a sub-quadratic transformer intended for generating text conditional on some long input context. The architecture, resemblant of the legacy encoder-decoder models, first compresses the context by chunking and applying a local encoder and temporal pooling. These compressed context representations are then prepended to the input of the decoder.

Authors proposed an approach to leverage existing encoders and decoders and combine them by means of fine tuning with adapters.

The proposal is simple and interesting, and the reported evaluations suggest it has potential to yield performant transformer variants. However, the manuscript has important issues at its current state even after back and forth with reviewers that prevent a recommendation for acceptance. To list a few examples:

- The presentation is lacking in various aspects and some important details are missing. For instance, section 3.2 doesn't provide details on training such as the datasets that are used to train the adapters and for how long they trained.
- Chunking approaches to yield sub-quadratic models was studied in the past and authors missed important references. One such example would be MegaByte and variants of it that were proposed more recently.
- The choice of baselines seems unfair sometimes since the proposed models were fine-tuned in such a way that targeted the reported evaluations, while not all baselines are fine-tuned. In fact, results would be more reliable if comparisons were carried out against baseline performances computed by others (e.g., using some leader board maintained by a third party) in addition to the baselines prepared by the authors.
- The motivation for the exact proposed architecture is unclear. There are several possible configurations that would leverage chunking and an encoder, and it's unclear why the authors picked the exact one reported in the paper.
- The main issue in my opinion is the choice of chunk sizes, which seem too small to yield significant improvements. The complexity under chunking $O(LC+L^2/C^2)$ will only be relevantly sub-quadratic if $C$ is large with respect to $L$. E.g., if one picks $C=L^{1/3}$, then we'd have $O(C^{4/3})$, but the chunk sizes in the experiments are much smaller than that. This also makes the performance results in Figure 4 a bit surprising since one would expect worse  scaling (almost as bad as quadratic) for such small chunks (512 chars).

All in all, while the approach is interesting, the manuscript needs further work prior to publication in terms of justifying the design choices better (either formally or via ablations), comparing to existing chunking-based approaches, and comparing agains results reported by others in addition to the authors implementations of baselines.

**Additional Comments On Reviewer Discussion:**

3/5 reviewers very briefly engaged with the authors in discussion, and most of them asked for more results on long context benchmarks. The authors did address those concerns by adding a significant amount of results, but important concerns remain as noted above.

---

### Decision · Program_Chairs · 2025-01-22

Reject